# Newly discovered *Synechococcus* sp. PCC 11901 is a robust cyanobacterial strain for high biomass production

Artur Włodarczyk[1,3 ✉], Tiago Toscano Selão [1,4], Birgitta Norling[1] & Peter J. Nixon [1,2 ✉]

Cyanobacteria, which use solar energy to convert carbon dioxide into biomass, are potential solar biorefineries for the sustainable production of chemicals and biofuels. However, yields obtained with current strains are still uncompetitive compared to existing heterotrophic production systems. Here we report the discovery and characterization of a new cyanobacterial strain, *Synechococcus* sp. PCC 11901, with promising features for green biotechnology. It is naturally transformable, has a short doubling time of ≈2 hours, grows at high light intensities and in a wide range of salinities and accumulates up to ≈33 g dry cell weight per litre when cultured in a shake-flask system using a modified growth medium — 1.7 to 3 times more than other strains tested under similar conditions. As a proof of principle, PCC 11901 engineered to produce free fatty acids yielded over 6 mM (1.5 g L$^{-1}$), an amount comparable to that achieved by similarly engineered heterotrophic organisms.

[1] School of Biological Sciences, Nanyang Technological University, Singapore, Singapore. [2] Sir Ernst Chain Building - Wolfson Laboratories, Department of Life Sciences, Imperial College London, S. Kensington Campus, London SW7 2AZ, UK. [3] Present address: Bondi Bio Pty Ltd, c/o Climate Change Cluster, University of Technology Sydney, 745 Harris Street, Ultimo, NSW 2007, Australia. [4] Present address: Department of Chemical and Environmental Engineering, University of Nottingham, University Park, Nottingham NG7 2RD, UK. ✉email: artur.wlodarczyk@bondi.bio; p.nixon@imperial.ac.uk

The current production of commodity chemicals from fossil fuels results in the release of greenhouse gases, such as $CO_2$, into the atmosphere. Given the concerns over the link between greenhouse gases and climate change and the limited abundance of cheap fossil fuels[1], alternative sustainable approaches need to be developed to produce carbon-based chemicals on an industrial scale. Yeast and bacteria are widely used biotechnology production platforms. However, their growth relies on the addition of carbohydrates to the growth medium leading to a 'food vs. fuel' dilemma, which will likely drive the price of most carbon feedstocks up, fuelled by the negative impact of global warming on food crops[2]. Cyanobacteria, especially marine strains, which can be cultivated in seawater not suitable for agricultural use or direct human consumption, have the potential to provide a completely sustainable solution[3–5]. These evolutionary ancestors of algal and plant chloroplasts are gram-negative prokaryotic oxyphotoautotrophs, able to convert $CO_2$ and inorganic sources of nitrogen, phosphorus and microelements into biomass[6].

Among cyanobacterial strains, the marine strain *Synechococcus* sp. PCC 7002[7] and several freshwater strains, namely *Synechocystis* sp. PCC 6803[8], *Synechococcus elongatus* PCC 7942[9] and, more recently, *Synechococcus elongatus* UTEX 2973[10], have become model organisms for both basic photosynthesis research as well as the photoautotrophic production of different chemicals such as bioplastics[11], biofuels (ethanol[12] and free fatty acids[13–15]) and specialised compounds like terpenoids[16,17]. However, the average yields are often low compared to heterotrophic microbes, partly due to slower growth and lower biomass accumulation.

In this work, we report the isolation and detailed characterisation of the novel marine cyanobacterial strain *Synechococcus* sp. PCC 11901 (hereafter PCC 11901) and demonstrate its potential for environmentally friendly biotechnological applications. PCC 11901 tolerates temperatures up to 43 °C, high light irradiances and salinities, with average doubling times ranging from 2 to 3 h in 1% (v/v) $CO_2$/air. We have also conducted a systematic parametric analysis to devise new culture media, termed MAD and MAD2, which enable cultures of PCC 11901 to produce the highest value of dry cell biomass compared to other cyanobacterial strains, when grown in a common shake-flask system. Lastly, we tested the ability of PCC 11901 to efficiently convert $CO_2$ into free fatty acids (FFA), a class of commodity chemicals used in a wide range of industries[18]. Although its current production from palm oil can be considered as renewable[19], growing world demand and extensive farming of palm oil trees in South East Asia are causing irreversible deforestation of primordial rainforests and degradation of the natural habitats in producing countries, raising questions on its sustainability[20]. Yields of FFA obtained with PCC 11901 reached 6.16 mM ($\approx$1.54 g L$^{-1}$) after 7 days of cultivation, a similar value to that attained by engineered *E. coli* with similar genetic manipulations[21] and several-fold greater[13–15] than yields previously achieved with other cyanobacteria.

## Results

**Strain isolation and characterisation.** As our primary goal was to isolate a fast-growing, preferably marine cyanobacterial strain that would not compete for freshwater resources and could tolerate a wide range of abiotic stresses, we collected seawater samples at a local floating fish farm located in the Johor river estuary in Singapore. Enrichment of water samples and consecutive re-streaking on solid medium resulted in the isolation of a fast-growing cyanobacterial colony. Inoculation into liquid AD7 medium, widely used to grow marine cyanobacteria[22,23], led to rapid growth but only after a long lag phase ($\approx$24 h), suggesting nutrient limitation. As most oceanic phytoplankton require cobalamin (vitamin $B_{12}$) for growth and its concentration in the seas and oceans varies from 0 to 3 pM[24,25], we supplemented cultures of the isolated xenic strain with 3 pM cobalamin, which alleviated the lag phase. Upon dilution plating of the culture on solid medium with cobalamin, small transparent bacterial colonies were found in the vicinity of the cyanobacterial colonies and both strains were further purified by streaking on a 1:1 mixture of AD7 and LB medium. In all, 16S rRNA sequencing and phylogenetic analyses revealed that the axenic cyanobacterial strain was a member of the *Synechococcales* group (now deposited in the Pasteur Culture Collection as *Synechococcus* sp. PCC 11901) and that the closest phylogenetic relatives to the companion heterotrophic bacterial strain belong to the marine *Thalassococcus* genus[26]. Both the axenic and xenic strains of PCC 11901 grew equally well in the presence of cobalamin, reaching an $OD_{730} \approx 23$ after 72 h (Fig. 1a). In contrast, growth of the axenic strain was severely inhibited in the absence of added cobalamin, with the observed residual growth possibly reflecting the retention of intracellular cobalamin or methionine reserves[27] in the inoculum (Fig. 1a). These results imply that PCC 11901 is auxotrophic, and that its growth depends on the availability of cobalamin in the growth medium.

Although the genome of the isolated *Thalassococcus* strain was not sequenced, another *Thalassococcus* sp. strain (SH-1) possesses cobalamin biosynthesis genes (GenBank accession

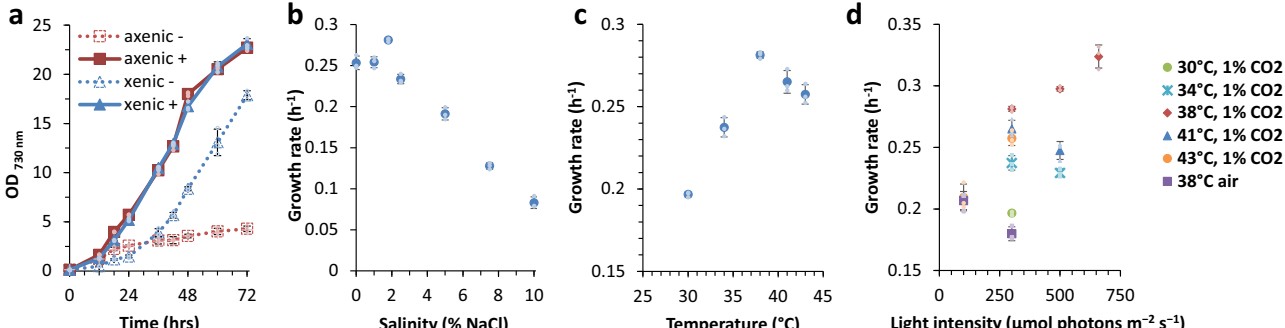

**Fig. 1 Growth analysis of *Synechococcus* sp. PCC 11901 strain. a** Axenic and xenic (contaminated with *Thalassococcus* sp.) strains grown in AD7 medium with (+) and without (−) addition of cobalamin (vitamin $B_{12}$). **b** Growth rate of axenic strain grown in AD7 medium with different (total) concentrations of sodium chloride (0–10%). For growth curves see Supplementary Fig. 3. Growth conditions for **a** and **b**: 38 °C, 225 rpm, 300 μmol photons m$^{-2}$ s$^{-1}$ light intensity (RGB LED 4:2:1 ratio) and 1% (v/v) $CO_2$. **c** Growth rates of the axenic strain cultured under different temperature conditions at 300 μmol photons m$^{-2}$ s$^{-1}$ light intensity and 1% (v/v) $CO_2$. **d** Growth rates of the axenic strain cultured under different light, temperature and $CO_2$ conditions. For all growth curves and doubling times see Supplementary Fig. 4 and Supplementary Table 1. Data points with error bars represent mean of $n = 3$ biological replicates ± standard deviation.

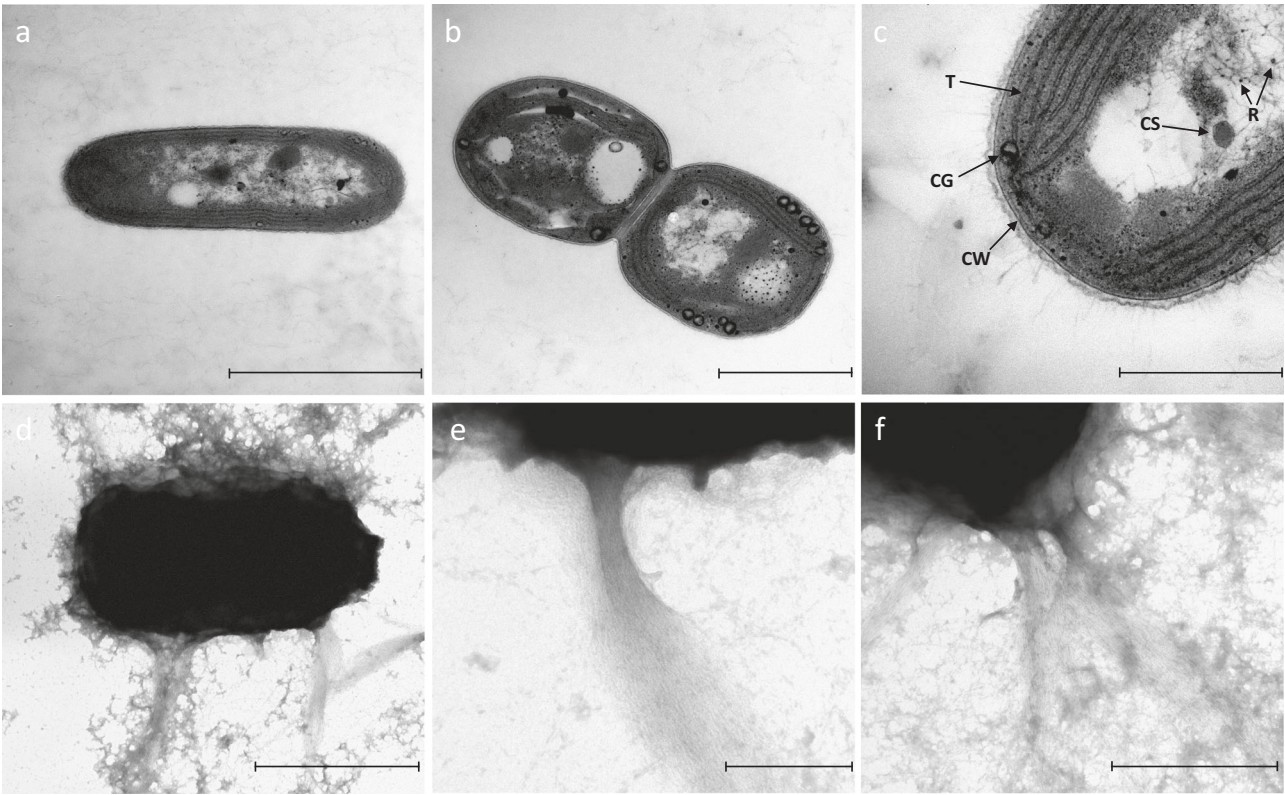

**Fig. 2 Transmission electron micrographs. a–c** Cross section of the cells. **d–f** Negatively stained cells. Scale: **a** 2000 nm, **b** 1000 nm, **c** 500 nm, **d** 1000 nm, **e** 200 nm, **f** 500 nm. CW cell wall, T thylakoids, CS carboxysome, R ribosome, CG cyanophycin granule.

number CP027665.1). Given the previous evidence for the mutualistic interaction between cobalamin-dependent micro-algae and bacteria[25], it is very likely that the *Thalassococcus* contaminant in the xenic culture could serve as a natural symbiotic partner for the PCC 11901 strain providing it with cobalamin, while consuming nutrients excreted by the cyanobacterium.

Similarly to PCC 7002[28], the PCC 11901 strain can grow mixotrophically in the presence of glycerol and photoheterotrophically in the presence of glycerol and the herbicide DCMU, which inhibits photosynthetic electron flow, although growth is impaired (Supplementary Fig. 1c). In contrast, very little or no growth was observed after one week of incubation with glucose and DCMU (Supplementary Fig. 1d).

Analysis of cell morphology revealed that PCC 11901, though unicellular, can also form short filaments of 2–6 cells, regardless of growth phase (Supplementary Fig. 2). Individual cells are elongated with sizes ranging from 1.5 to 3.5 μm in length and 1–1.5 μm in width. On average ($n = 45$), cells contain 4–6 concentric layers of thylakoids around the cytoplasm with visible convergence zones on the cell periphery (Fig. 2a–c). In negatively stained cells, long (1–1.5 μm) fibres (possibly pili or exopolysaccharide structures), similar to those seen previously in other cyanobacteria[29,30], were observed extending from the outer membrane (Fig. 2d–e).

**Growth performance under different conditions.** To test the tolerance of PCC 11901 to different salinities, growth was assessed in the presence of a wide range of NaCl concentrations, from 0 to 10% (w/v), where sea water is ≈3.5% NaCl (Fig. 1b, Supplementary Fig. 3a–c). PCC 11901 growth rates did not differ significantly (Supplementary data 1) in the 0–1.8% (w/v) NaCl concentration range (ANOVA, $p = 0.43$, $\eta^2 = 0.169$), with a

maximum of 0.28 $h^{-1}$ at 1.8% (w/v) NaCl. Higher NaCl concentrations had a negative impact on growth rate, though the strain was able to tolerate up to 10% (w/v) NaCl with a doubling time of 12 h (a similar growth rate to that of the halotolerant microalga *Dunaliella*[31]). While PCC 7002 can also grow in the presence of high salt (up to 9% (w/v) NaCl[32]), we found that in our hands the PCC 11901 strain exhibited higher salt tolerance (Supplementary Fig. 3d).

Different combinations of temperature, light intensity and $CO_2$ concentration were tested in order to investigate their effect on growth (Fig. 1c, d). As a reference, PCC 11901 was grown alongside two other fast-growing and high-temperature tolerant strains – UTEX 2973 and PCC 7002 (Supplementary Fig. 4). All strains grown in atmospheric $CO_2$ conditions and low light intensities (100 μmol photons $m^{-2}\,s^{-1}$) exhibited similar doubling times (Supplementary Table 1). At higher light the UTEX 2973 strain grew faster in atmospheric $CO_2$ conditions, though this result could be influenced by the presence of sodium bicarbonate in BG-11 medium (used to grow freshwater strains) and the absence of such a carbon source in AD7 medium (used to grow marine strains). The optimal growth condition for PCC 11901 was found to be 38 °C at 1% (v/v) $CO_2$ (Fig. 1c), with the shortest doubling time (2.14 ± 0.06 h) observed at a light intensity of 660 μmol photons $m^{-2}\,s^{-1}$ (Fig. 1d). The highest permissible growth temperature for PCC 11901 was 43 °C. Under our growth conditions, UTEX 2973 was the fastest growing strain (shortest doubling time of 1.93 ± 0.04 h at 41 °C, 500 μmol photons $m^{-2}\,s^{-1}$ and 1% (v/v) $CO_2$), though this advantage over PCC 11901 and 7002 strains becomes less apparent when grown for longer periods. After 24 h of cultivation UTEX 2973 reached lower $OD_{730}$ values than the other two strains and was unable to exceed $OD_{730} \approx 10$ after 4 days (Supplementary Fig. 5), an effect that was also previously observed[33]. In all light conditions tested, PCC 11901

accumulated more biomass after 4 days of growth (4.9 gDW L$^{-1}$) than PCC 7002 and UTEX 2973 (3.7 and 2.5 gDW L$^{-1}$ respectively; Supplementary Fig. 5a). However, all strains accumulated slightly less biomass at high light irradiances, an effect especially noticeable in the case of PCC 7002, which decreased from 3.1 to 2.6 gDW L$^{-1}$ at later time points (Supplementary Fig. 5b).

**Phylogenetic assignment and genome sequence analysis**. The genome of the PCC 11901 strain was sequenced using both Illumina and PacBio next-generation sequencing techniques, assembled and deposited in GenBank under accession number CP040360.1. Two sequences (one chromosome and one plasmid) were fully covered and circularised, and four shorter sequences (most likely plasmids) with sizes ranging from 10 to 98 kbp could not be closed. The complete genome size is 3,081,514 bp, similar in size to other phylogenetically related cyanobacterial strains, with an average GC content of 49.5%. The chromosome contains 42 tRNA, 6 rRNA, 4 ncRNA and 2943 gene sequences (3316 including plasmids). The distribution of all genetic elements on the genome is shown on 2 outer concentric circles corresponding to the directions of each of the two strands in Fig. 3.

The average nucleotide identity (ANI) analysis of the chromosome revealed 97.48% identity to *Synechococcus* sp. PCC 7117, 96.99% to both PCC 73109 and 8807, 96.76% to the commonly used PCC 7002 strain and 90.03% to PCC 7003. Complete genome sequences of the above mentioned strains were compared to the PCC 11901 strain genome using BLAST analysis in the CGView Server[34] tool. Rather unusually, there are several

major insertions in the genome of the PCC 11901, depicted by white gaps in the BLAST search scores shown in Fig. 3, the biggest of ~24 kbp, which are not found in other cyanobacterial genomes. Analysis of these fragments showed similarity to genes from other cyanobacteria outside the *Synechococcales* taxonomy order group, with a strong prevalence of genes encoding predicted glycosyltransferase genes (14 genes), ABC transporter components, transposases, toxin-antitoxin system components and an alcohol dehydrogenase (Supplementary Table 2).

In one of the endogenous plasmids (GenBank accession number: CP040361.1) we identified a gene encoding a 5-methyltetrahydropteroyltriglutamate-homocysteine S-methyltransferase homologue (MetE, GenPept: QCS51047.1) involved in cobalamin-independent methionine synthesis. Although most cyanobacteria are capable of synthesising the cobalamin or pseudocobalamin required by the cobalamin-dependent methionine synthase (MetH), some strains, such as PCC 7002, lack MetE, and are therefore strict cobalamin auxotrophs[27,35]. Surprisingly, despite carrying both variants of methionine synthase, PCC 11901 is dependent on added cobalamin for growth (Fig. 1a), suggesting potential mutations in either the *metE* gene itself or in the corresponding cobalamin riboswitch leading to either a loss of activity or low expression of this gene. Interestingly, genes encoding three enzymes of the cobalamin salvage pathway[36] - CobA (uroporphyrinogen-III C-methyltransferase, GenPept: QCS48576.1), CobQ (cobyric acid synthase, GenPept: QCS50783.1) and CobS (adenosylcobina-mide-GDP ribazoletransferase, GenPept: QCS48590.1) - can be found in the chromosome, suggesting that they could have been

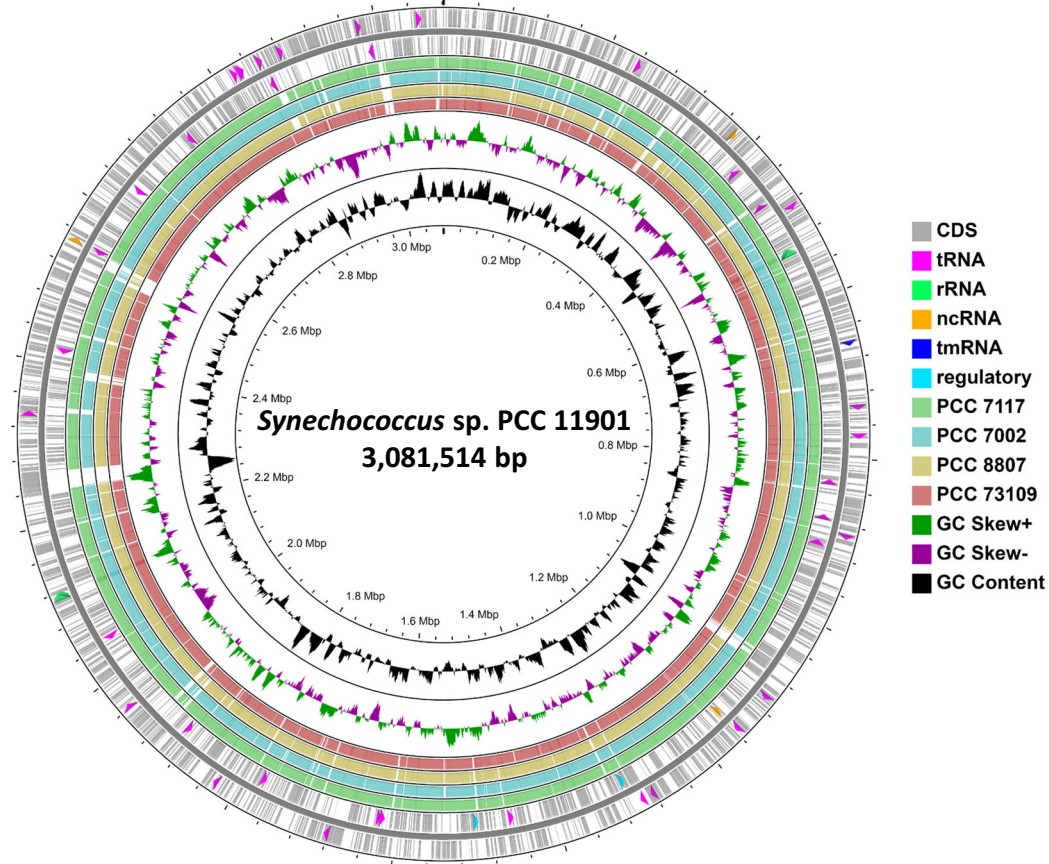

**Fig. 3 Circular diagram of the PCC 11901 genome sequence combined with the BLAST scores of four related cyanobacterial strains compared to PCC 11901 genome.** Tracks: CDS (grey), tRNA (pink), rRNA (green), ncRNA (orange), tmRNA (dark blue), regulatory elements (turquoise), GC Skew (above average: green, below average: purple) and GC content (black). BLAST scores: PCC 7117 (green), PCC 7002 (light blue), PCC 8807 (tan) and PCC 73109 (pink) compared to the PCC 11901 genome. Image created using CGView Server[BETA] online tool[34].

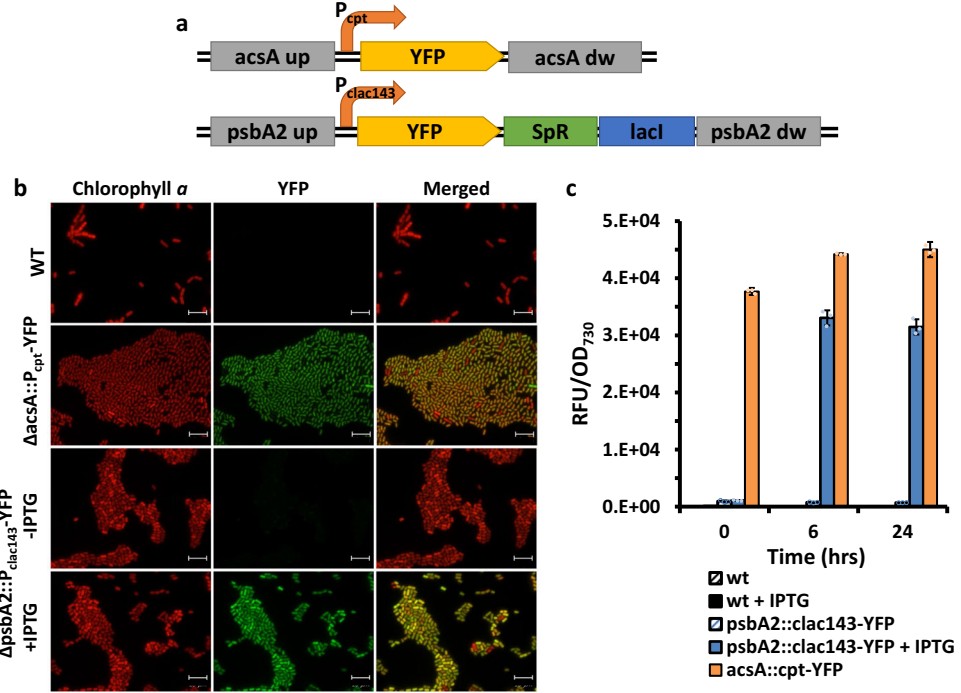

**Fig. 4 Usability of different molecular tools for the expression of heterologous proteins in *Synechococcus* sp. PCC 11901. a** Schematic illustration of constructs used for natural transformation and homologous recombination in the chromosome of 11901 strain. In the pSW036 vector the *yfp* gene was inserted between 700 bp flanking regions of the *acsA* gene and expression was controlled by the synthetic and constitutive $P_{cpt}$ promoter. In the pSW039 vector, the *yfp* gene (under control of the synthetic, inducible $P_{clac143}$ promoter) together with the *lacI* regulator and a spectinomycin-resistance cassette were inserted between 700 bp flanking regions of the *psbA2* gene. **b** Fluorescence microscopy images of cells transformed with YFP constructs. As a control, chlorophyll *a* autofluorescence of WT, Δ*acsA*::$P_{cpt}$-YFP and Δ*psbA2*::$P_{clac143}$-YFP is shown on the left side of the panel, and YFP fluorescence is shown in the middle panel. Δ*psbA2*::$P_{clac143}$-YFP strain was incubated for 24 h with/without addition of 1 mM IPTG prior to imaging. Scale 10 μm. **c** Strength and inducibility comparison of $P_{cpt}$ and $P_{clac143}$ promoters. Relative YFP fluorescence in relation to $OD_{730}$ was measured for cultures at $OD \approx 1$ ($T = 0$), after 6 and 24 h. The bars represent mean of $n = 3$ biological replicates ±standard deviation.

either acquired from other strains by horizontal gene transfer or, due to the availability of cobalamin in the local environment, the biosynthesis pathway could have been lost during evolution.

**Transformability and availability of molecular tools**. For a new strain to be useful for synthetic biology and metabolic engineering applications, it must be transformable and have molecular tools for controlled protein expression. Considering that PCC 11901 is a close relative of PCC 7002, we tested several molecular tools already established for PCC 7002. In one construct (Fig. 4a) the *yfp* gene was placed under control of the constitutive $P_{cpt}$ promoter (a truncated *cpcB* promoter from the PCC 6803 strain, functional in PCC 7002[37]) and inserted between flanking regions of the *acsA* (acetyl-CoA synthetase) gene, a method used previously to isolate markerless mutants of PCC 7002 using acrylic acid for counterselection[38]. In the second construct, the *yfp* gene and a spectinomycin-resistance cassette were cloned between flanking regions of the *psbA2* gene (encoding the D1 subunit of photosystem II) and YFP expression was controlled by the synthetic IPTG-inducible $P_{clac143}$ promoter[37]. PCC 11901 was then transformed by natural transformation and positive transformants were confirmed by colony PCR (Supplementary Fig. 6a, b). Fluorescence microscopy confirmed successful YFP expression in both constructs (Fig. 4b), with expression in the Δ*psbA2*::$P_{clac143}$-YFP strain detectable only after induction with IPTG. Though the targeted genome loci are different for these two constructs, the $P_{cpt}$ promoter appeared to be stronger than the induced $P_{clac143}$ promoter, as the relative fluorescence unit (RFU) per $OD_{730}$ ratio was higher, with YFP levels increasing 45-fold upon IPTG

induction in comparison to non-induced control cells (Fig. 4c). Transformation efficiency (assessed by disruption of the *fadD* locus using linear DNA containing an antibiotic resistance cassette gene) reached $1.03 \times 10^4$ CFU μg$^{-1}$, with all transformants fully segregated after the first round of restreaking (Supplementary Fig. 6g).

**Development of an improved growth medium**. Modifying the composition of the growth medium is a common approach to improve biomass and secondary metabolite production using heterotrophic microorganisms[39]. In contrast, most media routinely used to cultivate cyanobacteria were formulated over 40 years ago and are not optimised for high biomass production. Indeed, modification of the basic A$^+$ medium by Clark et al.[33] was shown to increase biomass production of PCC 7002 by ~2-fold (to 10 gDW L$^{-1}$) and in another report sulfur was identified as the major limiting nutrient for high-density growth of PCC 6803 using BG-11 medium[40].

Inspired by these reports we adopted a systematic approach to improve growth of PCC 11901 by independently optimising the levels of nitrate, phosphate and iron in AD7 medium (Fig. 5). According to Clark et al.[33], the 20 mM concentration of MgSO$_4$ in the regular A$^+$ (and AD7) medium is theoretically sufficient for a 10-fold biomass increase of the PCC 7002 strain and thus concentrations of Mg and S were not altered. Our experiments led to the development of a Modified AD7 medium (hereafter named MAD medium), containing 96 mM NaNO$_3$, 240 μM FeCl$_3$ and 1.2 mM phosphate. In contrast, medium A described by Clark et al. ("modified medium A" or "MMA" medium[33]) contains 122

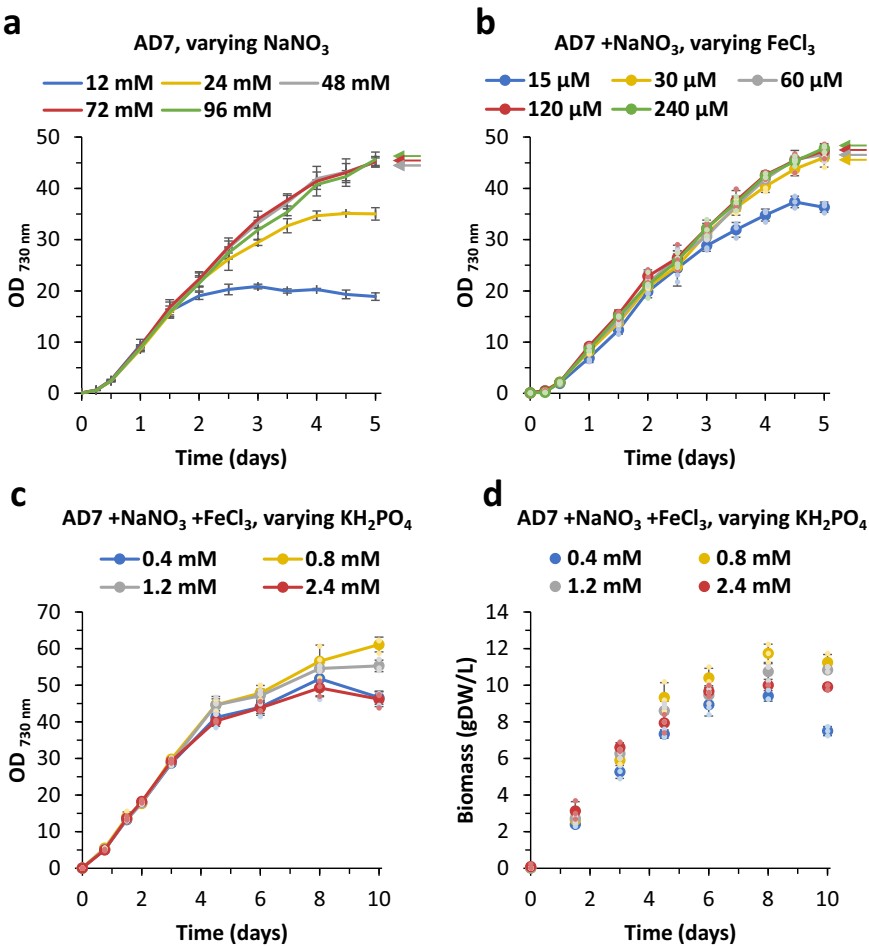

**Fig. 5 Effect of medium on growth of *Synechococcus* sp. PCC 11901.** Analysis of the effect of medium supplementation with increasing concentrations of sodium nitrate, iron (III) chloride and potassium dihydrogen phosphate on growth of PCC 11901. Growth conditions: 38 °C, 200 rpm, 300 µmol photons m$^{-2}$ s$^{-1}$ light intensity (RGB LED 4:2:1 ratio) and 1% $CO_2$. **a** AD7 medium with double the regular concentration of $FeCl_3$ (30 µM) was supplemented with a wide spectrum of $NaNO_3$ concentrations. **b** AD7 medium enriched with 96 mM $NaNO_3$ was supplemented with different concentrations of $FeCl_3$. **c**, **d** Modified AD7 medium with 96 mM $NaNO_3$ and 240 µM $FeCl_3$ was supplemented with increasing concentrations of $KH_2PO_4$ and tested for growth and biomass accumulation over 10 days. Data points with error bars represent mean of $n = 3$ biological replicates ± standard deviation.

mM $NaNO_3$, 1.1 mM ammonium iron citrate and a total of 31 mM phosphate (by adding 5.2 mM in a fed-batch scheme). We found that phosphate at levels >0.8–1.2 mM had an adverse effect on growth of PCC 11901 (Fig. 5d, Supplementary Fig. 7), though another report shows enhanced growth of PCC 7002 upon medium supplementation with 5 mM $KH_2PO_4$, 15 mM $Mg^{2+}$ and 5 mM $SO_4^{2-}$ [41]. The highest biomass accumulation was achieved after 8 days of cultivation in MAD medium (12.3 gDW L$^{-1}$) while in the case of MMA medium containing 5.2 mM phosphate, a maximum of 10.2 gDW/L was obtained in 8 days and 10.3 gDW L$^{-1}$ in 10 days when supplemented with 31 mM phosphate (Supplementary Fig. 7). This difference in biomass could reflect precipitation of iron, magnesium, calcium and other micronutrients as phosphate salts in MMA medium, thus lowering their bioavailability in basic pH.

**High biomass production in optimised medium.** To evaluate the general use of modified media for growing cyanobacteria (MAD for marine strains and a 5x concentrated BG-11 medium (5xBG) for freshwater strains), other commonly used strains – marine PCC 7002 and three freshwater strains PCC 7942, UTEX 2973 and PCC 6803 – were grown alongside PCC 11901 at 30 °C (the optimal temperatures for PCC 7942[42] and PCC 6803[43]) and with 1% (v/v) $CO_2$/air using a shake-flask system. PCC 11901

clearly outperformed all other model cyanobacteria, growing to an $OD_{730} = 101$ and accumulating a maximum of 18.3 gDW L$^{-1}$ of biomass, almost twice the biomass of PCC 7002 (9.3 gDW L$^{-1}$) under the same conditions (Fig. 6a–b). Among the freshwater cyanobacteria, PCC 6803 accumulated the highest biomass (6.9 gDW L$^{-1}$), followed by PCC 7942 and UTEX 2973 – both around 6.5 gDW L$^{-1}$. Regarding culture fitness, loss of the light-harvesting phycobilisome complex, which is symptomatic of general stress[44], was already apparent in the case of UTEX 2973 after 3–4 days of cultivation but less apparent in the PCC 7942 and 11901 strains, even after 10 days of cultivation (Fig. 6c, d). The observed differences in pellet size of the marine strains compared to freshwater strains may be explained by higher water retention and/or higher exopolysaccharide content (Fig. 6e).

Despite numerous efforts, we were unable to grow any of the freshwater cyanobacteria used in this study to biomass levels exceeding 7 gDW L$^{-1}$. Medium optimisation was more challenging, possibly due to a lower tolerance to higher ionic strength in the media by these strains. We tested five different media formulations for UTEX 2973, PCC 7942 and PCC 6803 (Supplementary Fig. 8, Supplementary Table 3). Supplementing BG-11 medium with even more nitrate, phosphate, ammonium iron (III) citrate and magnesium sulphate was met with limited success, as, notwithstanding a fast intial growth, cultures declined

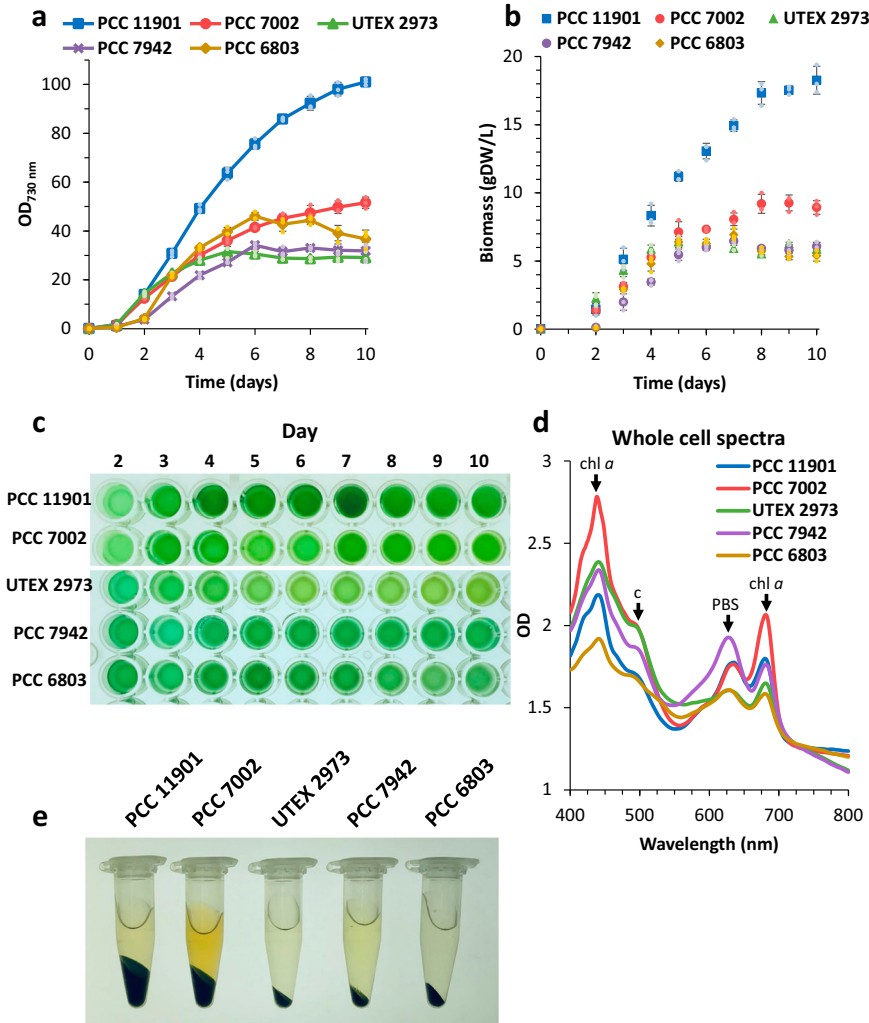

**Fig. 6 Comparison of growth performance and biomass accumulation of commonly used cyanobacteria cultured in optimised media.** All strains were grown in triplicates at 30 °C, 200 rpm shaking under continuous illumination using RGB LED 1:1:1 ratio. *Synechococcus* sp. PCC 11901 and 7002 were grown in the MAD medium at initially 150 μmol photons m$^{-2}$ s$^{-1}$, which was then increased to 750 μmol photons m$^{-2}$ s$^{-1}$ after 1 day. *Synechococcus elongatus* UTEX 2973 was grown in 5xBG medium under the same conditions. *Synechococcus elongatus* sp. PCC 7942 and *Synechocystis* sp. PCC 6803 were also grown in 5xBG medium, with light intensity gradually increased from 75 to 150 μmol photons m$^{-2}$ s$^{-1}$ (day 1) and eventually set to 750 μmol photons m$^{-2}$ s$^{-1}$ (day 2). **a**, **b** Growth curve and biomass accumulation comparison of all tested cyanobacterial strains. **c** Culture samples over 10 days of cultivation. Samples were diluted to demonstrate pigmentation and are not representative of the actual culture cell densities. **d** Whole-cell spectra of cultures harvested after 10 days of cultivation and diluted to the same OD$_{730}$. Highlighted peaks represent main absorption bands for: chlorophyll *a* (chl *a*) – λ = 440 and 680 nm, carotenoids (c) – λ = 490 nm and phycobilisomes (PBS) – λ = 620 nm. **e** Comparison of the cell pellet size from 1 mL of each culture harvested after 10 days of cultivation. Data points with error bars represent mean of *n* = 3 biological replicates ±standard deviation.

dramatically after a few days (Supplementary Fig. 7a–c). Cultures grown in modified MAD medium lacking sodium chloride did not perform well either (Supplementary Fig. 7d). In our hands, 5xBG containing a 5-fold higher concentrations of all nutrients (including trace metals) than the regular BG-11 was the most successful formulation (see Fig. 5 and Supplementary Fig. 8e, f) and to our surprise supplementing this medium with phosphate and nitrate concentrations to the levels in MAD medium (modified 5xBG, 5xBGM) led to complete bleaching of the PCC 6803 strain after 10 days (Supplementary Fig. 9a, b). Since the 5xBGM medium contains more nitrate and phophate than 5xBG, it is unlikely that the cultures bleached as a result of nutrient starvation.

In order to check whether higher CO$_2$ supplementation would improve biomass accumulation of marine strains, both PCC 11901 and PCC 7002 were cultured at 5% (v/v) CO$_2$ using MAD medium (Fig. 7). Although the effect of higher CO$_2$ on the growth

of PCC 11901 was minimal, PCC 7002 accumulated roughly 50% more biomass, reaching 15.8 gDW L$^{-1}$. To our surprise, further enrichment of the MAD medium with higher concentrations of N, P, Fe, trace minerals and cobalamin (termed "MAD2 medium") greatly improved biomass accumulation of PCC 11901, yielding 32.6 gDW L$^{-1}$, while a slight improvement was also observed for PCC 7002, which accumulated 19.2 gDW L$^{-1}$. These values are much greater than previous reports for cyanobacteria grown in batch culture. This large increase in biomass seems to be accounted for by an increase in cell size, as cell counts did not differ substantially from those of PCC 7002 (Supplementary Figs. 10–12). However, from the 5th day of culture in MAD2, 5% (v/v) CO$_2$ and 750 μmol photons m$^{-2}$ s$^{-1}$, PCC 11901 cells became considerably larger than those of PCC 7002 (Supplementary Fig. 10d), reaching a median cell length at day 10 of 3.01 ± 0.60 μm vs. 2.20 ± 0.41 μm for PCC 7002 (*n* = 102, *p* = 5.82 × 10$^{-23}$, see Supplementary Fig. 12c).

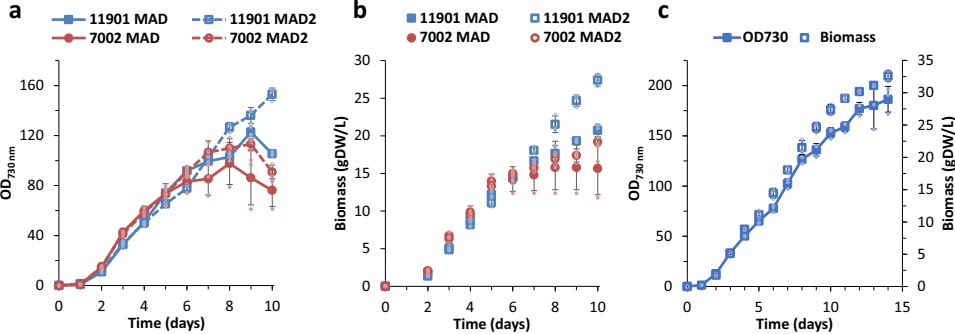

**Fig. 7 Comparison of growth performance and biomass accumulation of marine strains under presence of 5% (v/v) CO₂.** Strains were grown in triplicates at 30 °C, 250 rpm shaking under continuous illumination using RGB LED 1:1:1 ratio. *Synechococcus* sp. PCC 11901 and PCC 7002 were grown in the MAD and MAD2 medium at initially 150 μmol photons $m^{-2} s^{-1}$, which was then increased to 750 μmol photons $m^{-2} s^{-1}$ after 1 day. Data points with error bars represent mean of $n = 3$ biological replicates ± standard deviation. **a, b** Growth curve and biomass accumulation comparison of PCC 11901 and PCC 7002 strains in MAD and MAD2 medium. **c** Continued growth curve and biomass accumulation of PCC 11901 strain cultured in MAD2 medium for 14 days.

**Photoautotrophic production of free fatty acids**. Having demonstrated that the PCC 11901 strain is amenable to genetic manipulation, a proof-of-concept demonstration of the strain's biotechnological potential was designed, by modifying PCC 11901 for photoautotrophic production of FFA, which are relevant industrial feedstocks[18,45]. We inserted a truncated version of the *E. coli* thioesterase ('*tesA*) gene[13], codon-optimised for PCC 7002, under control of the inducible $P_{clac143}$ promoter[37], in the genomes of both PCC 11901 and PCC 7002, by simultaneously knocking-out the long-chain-fatty-acid-CoA ligase (*fadD*), to generate "11901 Δ*fadD*::*tesA*" and "7002 Δ*fadD*::*tesA*" strains (Fig. 8a). Knockout strains of the *fadD* gene alone (Δ*fadD*) and WT were used as production controls. In order to check if the MAD medium would further improve production yields, all strains were grown side-by-side using either AD7 or MAD medium.

Both PCC 11901 and PCC 7002 Δ*fadD*::*tesA* strains grown in AD7 medium excreted similar amounts of FFA (0.41 ± 0.07 and 0.34 ± 0.08 mM respectively) 3 days after induction (Fig. 8b), while the non-induced strains excreted <0.07 mM of FFA. The difference in productivity became much more apparent when the strains were grown in the MAD medium. The engineered PCC 11901 Δ*fadD*::*tesA* strain was able to grow faster than the PCC 7002 counterpart and 4 days after IPTG induction had already excreted 3.97 mM of FFA, nearly twice the amount of the PCC 7002 Δ*fadD*::*tesA* strain in the same time frame (2.01 mM) (Fig. 8c). Taking into account cell growth, the overall FFA productivity was higher for the PCC 11901 strain (0.12 mmol $L^{-1}$ per OD on day 5) compared to PCC 7002 (0.05 mmol $L^{-1}$ per OD and 0.08 mmol $L^{-1}$ per OD on day 5 and 7 respectively). Growth of both PCC 11901 and PCC 7002 strains was inhibited after 5–7 days of cultivation, possibly due to high concentrations of FFA in the culture medium, previously shown to have a negative effect on growth of other cyanobacteria[46]. The control PCC 7002 Δ*fadD* strain grown in MAD medium produced 0.1 mM of FFA, almost twice as much as the PCC 11901 Δ*fadD* control strain, with production levels in both WT being negligible, within statistical error (Fig. 8d). The large error bars in all FFA measurements are due to the low solubility of FFA in water, which makes them float at the air-liquid interface and adhere to culture vessel surfaces (Fig. 8f).

To test whether FFA production could be further improved by using different extraction methods and late-phase induction, the PCC 11901 Δ*fadD*::*tesA* strain was grown in MAD medium and FFA were extracted with hexane either directly from cell cultures or from the cell-free medium supernatant (Fig. 8e). For the first few days, differences between FFA concentration in the cell culture and

medium extracts were not substantial. However, after 7 days, samples extracted directly from the cell culture contained considerably more FFA (6.16 mM) than the ones from the supernatant.

The composition of produced FFA was evaluated by Gas Chromatography (GC) analysis in both PCC 11901 and PCC 7002 Δ*fadD*::*tesA* engineered strains (Supplementary Fig. 13), with no major differences detected between the two strains, despite their different final productivities. For the PCC 11901 Δ*fadD*::*tesA* strain, palmitic acid (C16:0) constitutes on average approximately 65% of the total FFA produced, followed by myristic acid (C14:0) – 23% and stearic acid (C18:0) – 9% (Fig. 8g).

## Discussion

In this work we report the discovery of the cyanobacterial strain *Synechococcus* sp. PCC 11901 from the Johor Strait (Singapore) and describe new media (MAD and MAD2) for high-density cultivation of this and similar marine cyanobacterial strains. PCC 11901 displays fast growth, high biomass accumulation and resistance to various abiotic stresses (such as high light and salinity), which are all important criteria for potential industrial uses of any cyanobacterial strain, especially when cultivated outdoors in a non-sterile environment and exposed to contamination that could potentially overgrow it. It was recently shown that engineering cyanobacteria to utilise unconventional phosphorus and nitrogen sources can dramatically reduce the risk of contamination[47]. Another common strategy for control of biological contamination is to increase the salinity of the growth medium. Here we show that PCC 11901 is an euryhaline strain that tolerates up to 10% (w/v) NaCl and can grow at high light intensities, up to at least 750 μmol photons $m^{-2} s^{-1}$ (the limit of our equipment). Although PCC 11901 tolerates high temperatures (up to 43 °C), its optimal growth temperature is within the 30–38 °C range, similar to the average temperature in its natural environment in Singapore (28–32 °C throughout the year). Though its cobalamin auxotrophy may be inconvenient for industrial cultivation, this can potentially be overcome by heterologous expression of a cobalamin-independent methionine synthase (MetE)[27]. Interestingly, PCC 11901 already possesses a *metE* gene in one of the endogenous plasmids, though it seems to be inactive or insufficiently active for rapid growth. This is possibly an evolutionarily recent mutation, most likely due to the ready availability of cobalamin in seawater[24].

In our hands, PCC 11901 shows a similar doubling time to PCC 7002 of 2.14 ± 0.06 h, marginally longer than that reported for UTEX 2973 (1.93 ± 0.04 h) (Supplementary Table 1). However, compared to both UTEX 2973 and PCC 7002, PCC 11901

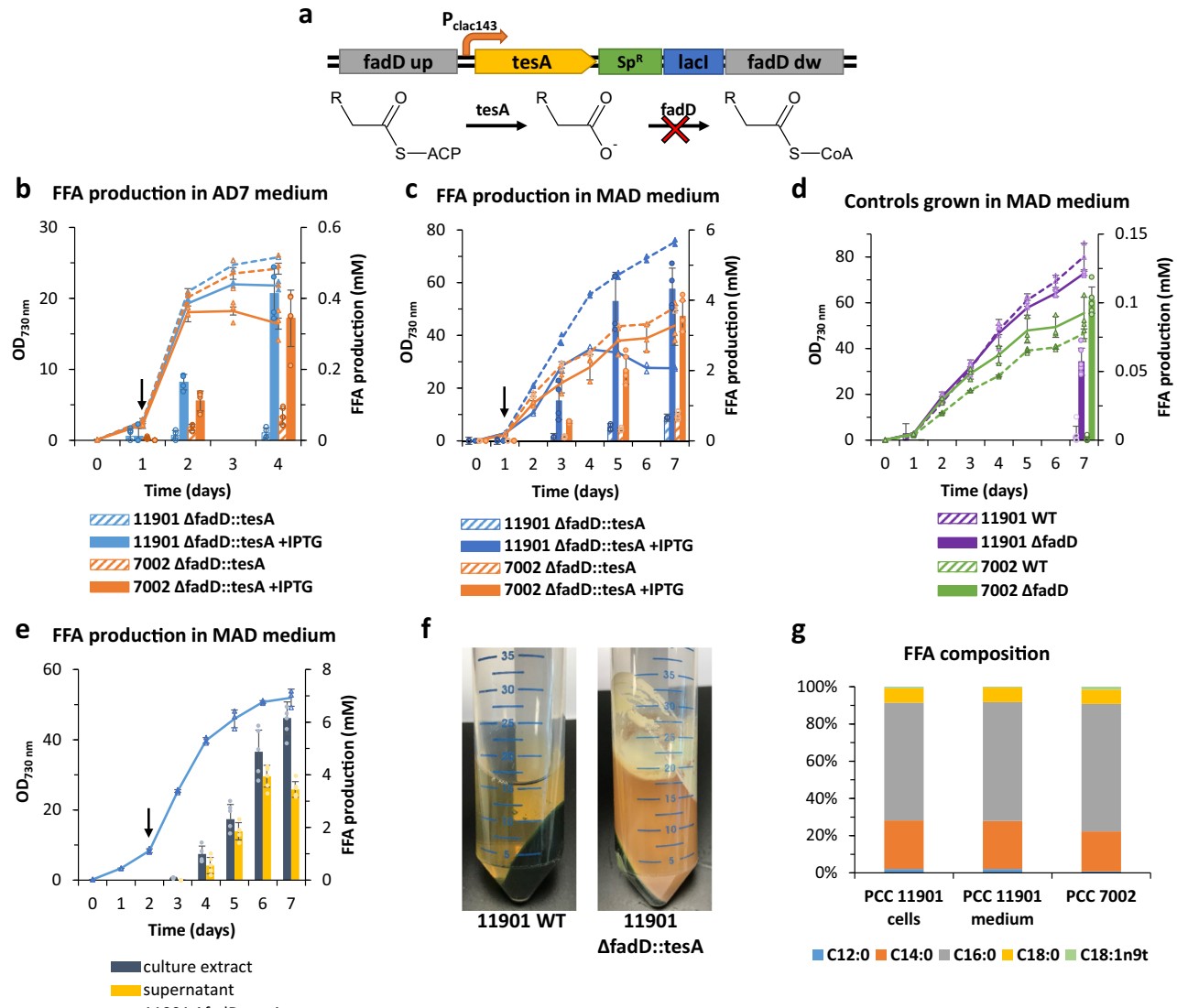

**Fig. 8 FFA production in engineered *Synechococcus* sp. PCC 11901 and 7002 strains. a** Schematic illustration of FFA production pathway. The truncated acyl-CoA thioesterase gene ('*tesA*), under control of $P_{clac143}$ inducible promoter, was inserted into the chromosome with simultaneous knockout of the endogenous long-chain-fatty-acid--CoA ligase (*fadD*). 'TesA converts fatty acyl-ACP to free fatty acid and its reconversion to fatty acyl-CoA is negated by knocking out the endogenous *fadD*. Comparison of the growth and FFA productivity of the engineered *Synechococcus* sp. PCC 11901 and PCC 7002 strains using regular AD7 (**b**) and MAD (**c**) medium. Filled and dashed lines indicate growth ($OD_{730}$ measurements) of the induced and non-induced cultures respectively, while the column bars correspond to the FFA production. Arrows indicate addition of IPTG inducer and light intensity increase. **d** Comparison of growth and FFA production by the WT (dashed lines and columns) and $\Delta fadD$ knockout control (filled lines and columns) strains. **e** Analysis of FFA production by the engineered PCC 11901 strain induced at $OD_{730} \approx 8$. FFA were extracted either directly from the culture or from the medium supernatant. For $OD_{730}$ measurements data points with error bars represent mean of $n = 3$ biological replicates ±standard deviation (for PCC 11901 $\Delta fadD::tesA$ strain $n = 2$ in panels **b** and **c** and $n = 3$ in panel **e**). FFA concentration as mean of $n = 6$ replicates (three biological and two technical) ±standard deviation (for 11901 $\Delta fadD::tesA$ strain $n = 4$ in panels **b** and **c** and $n = 6$ in panel **e**). **f** Detail of WT and engineered PCC 11901 strain cultures centrifuged after 7 days of cultivation. **g** Lipidomic profiles of the FFA from PCC 11901 and PCC 7002 FFA producer strains: C12:0 (lauric acid), C14:0 (myristic acid), C16:0 (palmitic acid), C18:0 (stearic acid) and C18:1n9t (elaidic acid).

cultivated in regular and optimised MAD medium can grow to much higher $OD_{730}$ and accumulate more biomass overall (Figs. 6a, b and 7; Supplementary Figs. 5 and 14), a feature that is often overlooked when comparing cyanobacterial strains for biotechnological applications.

Experimental testing of nutrient limitations allowed us to tap into a potential for ultra-high biomass accumulation. We show that PCC 11901 can grow to $OD_{730}$ of at least 186 and produce 32.6 g L$^{-1}$ of dry weight biomass (at a rate of $2.4 \pm 0.8$ gDW L$^{-1}$ per day ($\approx$100 mgDW h$^{-1}$, Fig. 7). When grown alongside PCC 7002, PCC 7942, PCC 6803 and UTEX 2973 strains, PCC 11901

accumulated 1.7–3 times more biomass than these strains despite using the same or similar nutrient concentrations (Table 1). Though PCC 7002 has been reported to accumulate up to $\approx$30 gDW L$^{-1}$ using specialised two-tier culture vessels[41], we were unable to grow this strain to such high densities in a shake-flask system. Possibly PCC 7002 has higher requirements for carbon and, due to the higher viscosity of its cultures (most likely caused by the release of intracellular metabolites into the medium), it might be difficult to sustain efficient gas exchange and achieve higher biomass yields when grown in a shake-flask system.

**Table 1 Comparison of the maximal biomass yields achieved for cyanobacterial strains under different growth conditions.**

| Strain | Growth conditions | Biomass (gDW/L) | Reference |
|---|---|---|---|
| *Synechococcus* sp. PCC 11901 | 1% (v/v) $CO_2$; batch culture in shake-flasks | 18.3 | This work |
| | 5% (v/v) $CO_2$; batch culture in shake-flasks | 32.6 | This work |
| *Synechococcus* sp. PCC 7002 | 1% (v/v) $CO_2$; batch culture in shake-flasks | 9.3 | This work |
| | 5% (v/v) $CO_2$; batch culture in shake-flasks | 19.2 | This work |
| | ≤10% (v/v) $CO_2$, 8.4 g/L $NaHCO_3$; fed-batch culture in two-tier vessel with permeable membrane | ≈30 | Ref. [41] |
| | 1% (v/v) $CO_2$; fed-batch culture in shake-flasks | 10 | Ref. [33] |
| *Synechocystis* sp. PCC 6803 | 1% (v/v) $CO_2$; batch culture in shake-flasks | 6.9 | This work |
| | ≤10% (v/v) $CO_2$, 8.4 g/L $NaHCO_3$; batch culture in two-tier vessel with permeable membrane | ≈11 | Ref. [41] |
| *Synechococcus elongatus* PCC 7942 | 1% (v/v) $CO_2$; batch culture in shake-flasks | 6.3 | This work |
| *Synechococcus elongatus* UTEX 2973 | 1% (v/v) $CO_2$; batch culture in shake-flasks | 6.5 | This work |

The PCC 11901 strain, in contrast to PCC 7002, seems to always accumulate biomass at similar rates, regardless of $CO_2$ concentration used (1 and 5%). That, in conjunction with other results, suggests that other cyanobacterial strains may require higher thresholds of N, P, C and microelements for high-biomass accumulation than PCC 11901, which is a clear advantage from an economical perspective.

It was previously shown that supplementation of BG-11 medium with 65 mM nitrate and 10 mM phosphate allows PCC 6803 to grow to high cell densities ($OD_{750} = 40$)[48] and to a maximum of ≈10 gDW $L^{-1}$ when less phosphate and nitrate were used[41]. Although such high values could not be achieved using our 5xBG medium recipe and shake-flask system, the difference in biomass yields compared to the two-tier vessel system utilised was not substantial. Therefore, it is possible that either these freshwater strains have intrinsic limitations, e.g. low tolerance to high concentration of inorganic salts, or they have a regulatory mechanism (e.g. quorum sensing) preventing them from growing further. While previous reports show that it is possible to grow *Arthrospira platensis* to biomass levels exceeding those in this study[49], this was achieved by daily replacement of enriched medium and use of irradiance levels far above peak solar intensity. However, it should be noted that *A. platensis* is not easily amenable to transformation, an important feature of the biorefinery concept.

Growing marine strains in both MAD and the previously described MMA medium[33] at higher temperatures (38 °C), led to a decline in biomass after 10 days of cultivation (Supplementary Fig. 7). In contrast, when grown at 30 °C, strains could sustain growth to higher densities, possibly due to the higher solubility of $CO_2$ at lower temperatures or a hitherto unknown regulatory mechanism.

From a biotechnological perspective, high biomass accumulation is an important feature directly translating into higher productivity of genetically engineered strains. Additionally, cyanobacterial biomass may be processed in biorefineries to extract food-grade all-natural pigments (phycocyanins, chlorophyll, carotenoids) or proteins and carbohydrates[50,51]. It was also shown that enzymatic lysates of cyanobacterial biomass contain high-levels of storage carbohydrates such as glycogen, which may be used as next-generation feedstock for yeast fermentation and production of bioethanol, addressing the "food vs. fuel" issue[5].

As a biotechnological chassis strain, PCC 11901 has several attractive traits: it is naturally transformable, facilitating genetic manipulation; synthetic biology tools already existing for the PCC 7002 strain[37,38] are compatible with it, facilitating an easy switch of metabolic engineering constructs to PCC 11901 and it shows high growth rates and high rates of biomass production.

As a proof-of-concept, we modified PCC 11901 to produce FFA, a biodiesel feedstock. FFA yields per litre of culture using MAD medium were found to be approximately 10-fold higher in comparison to using the regular AD7 medium under the same growth and induction conditions, even though the final $OD_{730}$ was only 1.5 times higher, demonstrating growth medium optimisation is a valid strategy to increase the production yields of FFA and possibly other biomolecules. However, further investigation will be necessary to understand the mechanism behind the observed changes to productivity.

The maximum FFA titre of 6.16 mM (≈1.54 g $L^{-1}$, 5 days after induction) achieved in this study is the highest reported to date for cyanobacteria, with values of 0.19 g $L^{-1}$ [15] and 0.13 g $L^{-1}$ [13] previously reported for PCC 6803 and PCC 7002, respectively, after 16–20 days of cultivation (see Table 2). It is also only 25% lower than the titre achieved by heterotrophic organisms with similar genetic modifications. For instance, introduction of different heterologous 'tesA genes in a ΔfadD strain of *E. coli* resulted in the production of ≈2 g $L^{-1}$ FFA 2 days post-induction[21]. Though these values are much lower than the highest ever reported FFA production titre of 50.2 g $L^{-1}$, using a heavily engineered *Rhodococcus opacus* strain, this latter system requires a high concentration of glucose[52]. It is likely that further engineering of the PCC 11901 strain and optimisation of growth conditions will further increase productivity. Given the reported FFA toxicity, continuous removal of the product from the growth medium using an organic overlay, as earlier demonstrated for PCC 7942 (up to 0.64 g $L^{-1}$ FFA)[14] and PCC 6803 (905 mg $L^{-1}$ of 1-octanol)[45], may also increase both final titres and overall cell viability.

In conclusion, we have shown that the newly isolated cyanobacterium *Synechococcus* sp. PCC 11901, in conjunction with the use of modified growth media, has the potential to become a relevant industrial biotechnology platform for the sustainable production of carbon-based molecules.

## Methods

**Strain isolation and purification.** PCC 11901 strain was isolated from the Johor Strait in the vicinity of Pulau Ubin island (Singapore, 1°25′17.7″N 103°57′20.6″E) and Johor river estuary (where the Johor river mixes with the seawater from the Singapore Strait and South China Sea). Water samples were collected at a depth of 20 cm below the water surface using sterile 50 mL Falcon tubes, mixed with AD7 growth medium (1:1) for marine cyanobacteria (for details see Supplementary Table 2) and transferred to culture flasks. Inocula were grown at 38 °C, 1% (v/v) $CO_2$, 160 rpm, under continuous illumination 200 µmol photons $m^{-2} s^{-1}$ for several days and subcultured three times. 50 µL of $2 \times 10^5$ culture dilutions were plated on solid growth medium. To avoid exposing cells to potential mutagens (such as antibiotics commonly used to remove bacterial contamination), the enriched cultures were dilution-plated and resulting single colonies were consecutively re-streaked on solid AD7 medium. Single colonies were subsequently restreaked and xenic strains were additionally restreaked on plates containing

**Table 2 FFA production yields achieved by engineered photoautotrophic and heterotrophic hosts having similar modifications.**

| Host | Modification | Production time | Yields | Reference |
|---|---|---|---|---|
| PCC 6803 | Δslr1609::P$_{psbA2}$ 'tesAΔ(slr1993-slr1994):: P$_{cpc}$accBC P$_{rbc}$-accDA Δsll1951::P$_{psbA2}$ Uc fatB1 P$_{rbc}$ Ch fatB2Δ(slr2001-slr2002):: P$_{psbA2}$ Ch fatB2Δslr1710::P$_{psbA2}$ Cc fatB1Δslr2132::P$_{trc}$ tesA137 | 16 days | 197 mg/L | [15] |
| PCC 7002 | ΔfadD, 'tesA, P$_{psbAI}$-rbcLS | 20 days | 130 mg/L | [13] |
| PCC 7942 | Δaas, P$_{nirA}$-tesA, isopropyl mirystate overlay | 20 days | 0.64 g/L | [14] |
| PCC 7002 | ΔfadD::P$_{clac143}$-'tesA * | 7 days | 885 mg/L | This work |
| PCC 11901 | ΔfadD:: P$_{clac143}$-'tesA * | 5–7 days | 1 -1.54 g/L | This work |
| E. coli | ΔfadD::tesA * | 2 days | 2 g/L | [21] |
| E. coli | ΔfadE, 'tesA, fadR | 3 days | 5.2 g/L | [59] |

*Indicates a comparable genetic modification.

cobalamin in order to remove the bacterial contaminants. To confirm the purity of the new isolate, the strain was restreaked on an LB agar plate and incubated for several days at 38 °C. Both the isolated strain and bacterial contaminant were identified as previously described by amplification of 16S rRNA fragments and Sanger sequencing using primer pairs CYA361f, CYA785r[53] and 27f, 1492r[54].

**Strains and growth conditions**. Newly isolated *Synechococcus* sp. PCC 11901 and PCC 7002 (obtained from the Pasteur Culture Collection) were grown photo-autotrophically in medium AD7 (medium A[22] with D7[23] micronutrients lacking NaVO$_3$). Solid medium was prepared by adding 1.2% (w/v) Bacto-Agar (BD Diagnostics) and 1 g L$^{-1}$ sodium thiosulfate.

Cyanobacterial natural transformation was performed by adding 0.5 μg of respective plasmids to 1 mL of culture at OD$_{730}$ ≈ 0.5 and incubating overnight under the same conditions. Cells were transferred onto solid medium supplemented with either 20 μg mL$^{-1}$ spectinomycin (and 10 mM glycerol when integrating into the *psbA* (NCBI: QCS50639.1) locus), 50 μg mL$^{-1}$ kanamycin or 100 μM acrylic acid, as required, grown for 4 days and restreaked twice to ensure complete segregation.

All cloning steps were performed using supercompetent *E. coli* cells (Stellar, TaKaRa), grown in LB medium at 37 °C, supplemented with appropriate antibiotics (100 μg mL$^{-1}$ carbenicillin or 50 μg mL$^{-1}$ spectinomycin). Transformation efficiency of PCC 11901 was assessed by transforming 1 mL of cells at OD$_{730}$ = 0.5 with 0.5 μg of ΔfadD::Kan$^R$ PCR product amplified with M13F-pUC (-40) and M13R-pUC (-26) primers from the pSW071 vector template. Total number of colony forming units (CFU) was then divided by the amount of DNA used for transformation.

For starter cultures, the freshwater cyanobacteria *Synechococcus elongatus* sp. UTEX 2973 (a kind gift from Prof. Poul Erik Jensen), *Synechococcus elongatus* sp. PCC 7942 (a gift from Prof. Adrian Fisher, Cambridge University) and *Synechocystis* sp. PCC 6803 (obtained from the Pasteur Culture Collection) were grown in medium BG-11[22].

For all growth curve experiments using regular AD7 and BG-11 media, 25 mL cultures adjusted to OD$_{730}$ ≈ 0.1 were grown in Erlenmeyer baffled flasks, in triplicates, and shaken at 225 rpm using an NB-101SRC orbital shaker (N-Biotek, Korea) in a 740-FHC LED incubator (HiPoint Corporation, Taiwan) (Supplementary Fig. 15), under constant illumination using an RGB LED Z4 panel set with R:G:B ratios either 4:2:1 (for cultures grown at 100, 300 and 500 μmol photons m$^{-2}$ s$^{-1}$ light intensities) or 1:0:1 (for cultures grown at 660 μmol photons m$^{-2}$ s$^{-1}$ light intensity). Lighting irradiances were measured at the culture surface using a quantum flux meter (Apogee Instruments, model MQ-500), and temperature and CO$_2$ conditions were set to values specified in Supplementary Table 1, as required. Cells growth was monitored by measuring the optical density at 730 nm (OD$_{730}$) in a 1-cm light path with a Cary 300Bio (Varian) spectrophotometer. Doubling times were calculated as the mean value within the logarithmic range only and statistically significant differences between them were ascertained using the data analysis package in Excel for Office 365 (version 16.0) to run an ANOVA analysis and two-tailed *t*-tests (see Supplementary data 1).

For the cobalamin auxotrophy experiment, a starter culture of the axenic strain was grown in medium AD7 supplemented with cobalamin, but the inoculum was washed three times with AD7 lacking cobalamin in order to remove any remaining cobalamin from the medium.

To facilitate acclimation to high salt concentrations cultures were incubated in low light for 1 week. Once the cultures were acclimated to the respective conditions, biological triplicates were inoculated to starting OD$_{730}$ of 0.1 and grown at 38 °C and 300 μmol photons m$^{-2}$ s$^{-1}$ light intensity.

For the biomass evaluation in the medium optimisation experiments 0.4–1 mL of cell cultures were filtered onto pre-dried and pre-weighed glass microfiber filters (47 mm diameter, 1 μm pore size, GE Healthcare, Cat. No. 1822-047), washed three times with MilliQ water in order to remove salt remains and dried for 24 hours at 65 °C (at which point the measured masses did not differ by more than ±0.0001 g).

For the comparison of all cyanobacterial strains 3 × 33 mL of either MAD or 5xBG optimised media (for details see Supplementary Table 3) media were inoculated with cells to a starting OD$_{730}$ of 0.1 and grown with shaking at 200 rpm, 30 °C, 1% (v/v) CO$_2$, with RGB LED ratio of 1:1:1. For growth of PCC 11901, PCC 7002 and UTEX 2973, the initial light intensity was set to 150 μmol photons m$^{-2}$ s$^{-1}$ increased after 1 day to 750 μmol photons m$^{-2}$ s$^{-1}$. In the case of PCC 7942 and PCC 6803, due to their lower light tolerance, the initial light intensity was set to 75 μmol photons m$^{-2}$ s$^{-1}$, changed to 150 μmol photons m$^{-2}$ s$^{-1}$ after 1 day and further increased to 750 μmol photons m$^{-2}$ s$^{-1}$ on the next day. To compensate for water evaporation 700 μL of sterile MilliQ water was added to the cultures on a daily basis (measured by weight difference). For the assessment of biomass accumulation at 5% (v/v) CO$_2$ using both MAD and MAD2 media, 1–2 μL of sterile Antifoam 204 (Sigma-Aldrich) was added to PCC 11901 and PCC 7002 cultures to facilitate gas exchange. Due to the difficulty in measuring the dry weight of UTEX 2973 cells by filtration (as they pass through the microfiber filters), an alternative method for determining the dry biomass of all strains was employed. In total, 0.5–1 mL cultures were transferred into pre-weighed Eppendorf tubes and centrifuged at 20,000×g for 2 min. All cell pellets were washed three times with 1 mL of MilliQ water in order to dissolve and remove remaining salt precipitates and dried at 65 °C for 24 h.

**Genome sequencing**. Genomic DNA was extracted using Quick-DNA Fungal/ Bacterial Kit (Zymo Research), concentrated and analysed by PacBio and Illumina MiSeq next-generation sequencing. Illumina MiSeq sequencing library construction was performed according to Illumina's TruSeq Nano DNA Sample Preparation protocol. Libraries were pooled at equimolar concentrations and sequenced on the Illumina MiSeq platform at a read-length of 300 bp paired-end.

PacBio run library preparation was performed using SMRTbell template prep kit 1.0 (Pacific Biosciences, USA) followed by single-molecule real-time sequencing on the PacBio RS II platform. The PacBio reads were assembled using HGAP3 assembler[55], and then polished with Quiver within the SMRT Analysis v2.3.0 protocol. Polished assembly contigs were then circularised and re-oriented with Circlator 1.1.4[56]. Assembled genomes were corrected by mapping 3 MiSeq Illumina paired reads to the reference sequences using the Geneious Prime® 2019.1.3 software (Biomatters Ltd.) and consensus sequences were exported and deposited to GenBank. Average nucleotide identity (ANI) analysis was performed using OrthoANIu algorithm[57]. Circular diagrams of the PCC 11901 genome sequence and BLAST analysis were created using CGView Server$^{BETA}$ online tool[34].

**Glycerol and glucose adaptation**. AD7 medium supplemented with 10 mM glycerol or 0.15% (w/v) glucose was inoculated with a single colony of PCC 11901 and grown for 1 week at a low light intensity (50 μmol photons m$^{-2}$ s$^{-1}$), 38 °C, 1% (v/v) CO$_2$. Growth was observed after a few days and strains were restreaked on AD7 agar plates with either 10 mM glycerol or 0.15% (w/v) glucose. For the photoheterotrophy assay glycerol and glucose adapted strains were grown to OD$_{730}$ ≈ 5 and 15 μL dilution series were transferred onto AD7 agar plates containing different combinations of 10 μM DCMU, 10 mM glycerol or 0.15% (w/v) glucose. Plates were dried and incubated for 7 days at 50 μmol photons m$^{-2}$ s$^{-1}$, 38 °C, 1% (v/v) CO$_2$.

**Cloning of constructs**. All fragments needed for cloning of pSW036, pSW039, pSW040, pSW068, pSZT025, pSW071 and pSW072 vectors were amplified using Q5® High-Fidelity DNA Polymerase (New England Biolabs) according to the manufacturer's protocol. All primers and templates used for PCR amplification are listed in Supplementary Table 4. PCR products were incubated overnight with DpnI restriction enzyme and purified using an EZ-10 Spin Column PCR Products Purification Kit (BioBasic). Fragments were then ligated using NEBuilder® HiFi DNA Assembly (New England Biolabs) according to manufacturer's protocol and transformed into competent *E. coli* cells (Stellar, TaKaRa). A synthetic version of the *E. coli* thioesterase gene tesA lacking its cognate signal sequence ('tesA) was codon-optimised for Syn7002 (by GenScript, Hong Kong, Ltd). pAcsA_cpt_YFP and pAcsA_cLac143_YFP[37] vector templates for PCR were a kind gift from Prof.

Brian Pfleger, University of Wisconsin-Madison, USA and pDF-trc was a kind gift from Prof. Patrik Jones, Imperial College London, UK.

**YFP fluorescence measurements**. Triplicates of WT, $\Delta acsA$::$P_{cpt}$-YFP and $\Delta psbA2$::$P_{clac143}$-YFP strains were grown in regular liquid AD7 medium to $OD_{730} \approx 1$ (time, $T = 0$), and, if required, induced with 1 mM IPTG. Whole cell fluorescence was measured by transferring 150 μL of cultures, collected at $T = 0$, after 6 and 24 h into 96-well black clear bottom plates and measured with a Hidex Sense plate reader, using 485/10 and 535/20 nm filters for excitation and emission, respectively. Relative fluorescence was normalised to $OD_{730}$ (RFU/$OD_{730}$). All measurements were performed in triplicate.

**Fluorescence microscopy**. Liquid cultures of WT, $\Delta acsA$::$P_{cpt}$-YFP and $\Delta psbA2$::$P_{clac143}$-YFP strains were harvested at $OD_{730} \approx 1$ by centrifugation, concentrated and transferred onto 0.5% agarose pads placed on microscopy slides. Once dry, pads were covered with a coverslip. Fluorescence microscopy was performed using an Axio Observer Z1 (Zeiss) inverted fluorescence microscope with EC Plan-Neofluar 100×/1.30 Oil Ph 3 objective and immersion oil (total magnification 1000×). To measure chlorophyll autofluorescence and YFP fluorescence, excitation/emission wavelengths were 577/603 and 525/538 nm respectively and images were exposed for 130 ms. For cell size estimation, a total of 102 cells from each strain were measured using the ZEN software (Carl Zeiss, version 2.3) measuring tool, in three different fields of observation, at the same level of magnification. Population cell lengths were analysed using ANOVA and a post-hoc two-tailed $t$-test (as described above) and were found to be significantly different under the conditions tested (ANOVA, $p = 5 \times 10^{-23}$, $\eta^2 = 0.621$).

**Transmission electron microscopy**. Cultures were harvested by centrifugation, fixed for 1 h at room temperature with 4% (w/v) glutaraldehyde in 100 mM phosphate buffer (pH 7.3) and washed 3× with 100 mM phosphate buffer. After embedding in 2% (w/v) low-gelling-temperature agarose, samples were cut in 1–2 mm cubic blocks, and post-fixed with 1% (w/v) osmium tetroxide in distilled water for 1 h. Samples were washed twice with distilled water, and dehydrated through a graded ethanol series (1 × 15 min 50%, 1 × 15 min 70%, 1 × 15 min 90% and 3 × 20 min 100%). Two 5 min washes with acetone were performed prior to infiltration with Araldite for 1 h and with fresh Araldite overnight. Polymerisation was achieved by incubation at 60–65 °C for 48 h. Ultrathin sections were cut with a diamond knife at a Reichert Ultracut E microtome and collected on uncoated 300-hexagonal mesh copper grids (Agar Scientific). High contrast was obtained by post-staining with saturated aqueous uranyl acetate and Reynolds lead citrate for 4 min each.

Negative staining was performed on 300-mesh copper carbon supports grids (Agar Scientific) that were previously rendered hydrophilic by glow discharge (Easy-Glow, Ted Pella). Glutaraldehyde fixed bacteria were adsorbed to TEM grids by direct application of 5 μl of the suspension for 1 min and stained by floating the loaded grid onto a drop of 1% (w/v) uranyl acetate for 20 s. The grids were examined in a JEOL JEM-1230 transmission electron microscope at an accelerating potential of 80 kV.

**Free fatty acid production assay**. The PCC 11901 strain was transformed with the pSW068 and pSW071 vectors generating 11901 $\Delta fadD$::$tesA$ (production) and $\Delta fadD$ (control) strains respectively. PCC 7002 was transformed with pSZT025 and pSW072 to generate 7002 $\Delta fadD$::$tesA$ (production) and $\Delta fadD$ (control) strains. Successful transformants were screened for complete segregation by colony PCR (Supplementary Fig. 6) using primers listed in Supplementary Table 5. A large number of the screened colonies for the PCC 11901 strain did not carry the designed insert, suggesting that kanamycin may not be the antibiotic of choice for selection in this strain, as it exhibits partial resistance to kanamycin at low concentrations (below 50 μg mL$^{-1}$).

All engineered and control strains were grown in 33 mL cultures, in biological triplicates, (except for PCC 11901 $\Delta fadD$::$tesA$ grown in biological duplicates) using either basic AD7 or MAD medium. Cultures were inoculated with cells to a starting $OD_{730}$ of 0.1 and grown with shaking at 200 rpm, 30 °C, with RGB LED ratio 1:1:1 at 150 μmol photons m$^{-2}$ s$^{-1}$ light intensity. After 1 day, cultures were induced with 1 mM IPTG and the light was increased to 750 μmol photons m$^{-2}$ s$^{-1}$. In all, 1 mL medium aliquots were collected for both $OD_{730}$ and FFA quantification. For FFA quantification, cell cultures were centrifuged for 2 min at 20,000 × g and medium supernatants were carefully collected to avoid any disruption of the cell pellets. FFA were quantified in technical duplicates using the EZScreen™ Free Fatty Acid Colorimetric Assay Kit (384-well) (BioVision, USA) according to the manufacturers' protocol.

For GC analysis both cell extract and medium supernatant were acidified with 1 M HCl to pH ≈ 2 in order to protonate FFA and facilitate extraction. Samples were extracted with $n$-hexane, evaporated and dried using a centrifuge vacuum concentrator. Dried samples were resuspended in 100 μL of a 2:1 chloroform: methanol mixture and aliquots were transferred on to TLC Silica gel 60 plates (Merck, Germany). Plates were resolved for ~30 min in hexane, diethyl ether, formate solvent mixture at a 70:30:2 ratio. Sample preparation and the GC analysis was performed as previously described[58]. Briefly, after running and drying, plates

were sprayed with a 0.05% primuline solution and FFA bands visualised using a UV lamp. Individual FFA bands were then scraped from the TLC plates onto 2 mL tubes, 100 μL of 1 M C15:0 (pentadecanoic acid) internal standard were added to each tube and FFA were derivatized to fatty acid methyl esters (FAME) by incubation with 300 μl of 1.25 M HCl-methanol for 1 h at 80 °C. FAMEs were then extracted three times with 1 mL hexane, extracts were combined, dried under nitrogen and resuspended in 100 μL hexane. FAMEs were separated on a GC-2014 gas chromatographer equipped with a flame-ionization detector (Shimadzu, Kyoto, Japan), using an ULBON HR-SS-10 50 m × 0.25 mm column (Shinwa, Tokyo, Japan). Individual components were identified by comparison to the retention times of the Supelco 37 component FAME mix (Sigma-Aldrich, St Louis) and data were normalised using the internal C15:0 standard.

**Flow cytometry analysis**. Samples from PCC 7002 and PCC 11901 cultures, grown in MAD2 and 5% (v/v) $CO_2$, under 700 μmol photons m$^{-2}$ s$^{-1}$ were analysed using a Fortessa X20 flow cytometer at a consistent rate of 3000 events/s, as previously described[47]. Cell counts in duplicate samples were derived using the BD FACS Diva Software (v. 8.0) and back-calculated based on the dilution utilised. Median forward scatter measurements (derived using Flowing Software, version 2.5.1) for both strains were used to follow variations in cell size during growth.

**Statistics and reproducibility**. Statistical tests and post-hoc analyses were performed in MS Office Excel using the integrated Data Analysis plugin. To compare different conditions in Fig. 1b (within 0–2.5% NaCl range) and size difference of PCC 11901 and PCC 7002 strains in Supplementary Fig. 12 two-tailed $t$-test analysis and one-way analysis of variation (ANOVA) were applied to the raw datasets. Wherever applicable, a Bonferroni correction was used to calculate $p$-value thresholds. Unless otherwise specified, a total of $n = 3$ biological replicates were used to calculate average and standard deviation. Due to low solubility of fatty acids in aqueous buffers, two technical replicates were measured for each biological replicate, to minimise the error rate and ensure good data reproducibility throughout the experiment.

**Reporting summary**. Further information on research design is available in the Nature Research Reporting Summary linked to this article.

## Data availability

*Synechococcus* sp. PCC 11901 strain has been deposited in the Pasteur Culture Collection (Paris, France) and is available for purchasing upon request. The complete genome sequence of PCC 11901 strain was assembled and deposited in GenBank under accession number CP040360.1. Circularised plasmid was deposited under accession number CP040361.1 and remaining sequences that couldn't be circularised were deposited under accession numbers: CP040356.1, CP040357.1, CP040358.1 and CP040359.1. Plasmids used for cyanobacterial transformation were deposited in Addgene: pSW036 (ID: 140034), pSW039 (ID: 140035), pSZT025 (ID: 140033), pSW068 (ID: 140036) and pSW071 (ID: 140037). Raw data for all growth experiments has been deposited in Figshare: https://doi.org/10.6084/m9.figshare.11917893.v1.

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

## Acknowledgements

This work was supported by NTU grants M4080306 to BN and M4081714 to P.J.N. We would like to thank Daniela Moses (SCELSE, NTU) for assistance with whole genome sequencing reactions, Anthony Wong (SCELSE, NTU) and the SBS Microscopy facility for assistance with fluorescence microscopy imaging, Giulia Mastroianni (School of Biological and Chemical Sciences, Queen Mary University of London) for TEM sample preparation and analysis, the SBS Flow Cytometry facility and Prof. Adriana Lopes dos Santos (ASE, NTU) for assistance with flow cytometry data analysis and Michael Voigtmann (Wintershine Pte. Ltd.) for providing access to the sample collection site. The authors are also grateful to Prof. Bertil Andersson for his constant support and encouragement and his pivotal role in establishing the CyanoSynBio@NTU laboratory.

## Author contributions

A.W. and T.T.S. performed all experiments. Conception, design and data analysis were carried out by A.W., T.T.S. and P.J.N. Manuscript was written by A.W., T.T.S. and P.J.N.; B.N. critically reviewed the manuscript.

## Competing interests

The authors declare no competing interests.
