## [Peer Review File · Communications Biology]

Reviewers' comments:

Reviewer #1 (Remarks to the Author):

This is a landmark paper and a major advance in developing cyanobacteria as a biotechnology platform. While 'fast growing' cyanobacterial strains, such as *Synechococcus* UTEX 2973, have previously been reported, results from these papers show growth only at early log phase, which is not useful for biotechnology. I commend the authors for performing these studies properly by demonstrating fast growth to high cell concentrations and biomass, and for their laborious experiments on media optimisation. I am happy to support publication of this paper once the following comments have been addressed.

One big issue is how do the authors know that the higher biomass and OD of 11901 compared to the other species is due to faster growth and not differences in cell properties? It could be due to differences in cell size or in their optical properties of the cells (which could be accumulating storage compounds like polyhydroxybutyrate that change these). Were cell counts taken? Can the authors confirm there are more cells in the 11901 cultures at later growth stages?

Minor comments

The supplementary labels are not in the correct numerical order within the text (i.e. Fig. S1C is first in the text and fig. S1a or b are not mentioned). Please correct this and check the manuscript thoroughly.

Fig S5. It is hard to see both the biomass and growth measurements when presented in the same graph. Can this be split into separate graphs.

Line 167. It looks like some of the strains accumulated similar biomass at both light intensities although it is hard to tell since the graphs are so crowded. Please add numbers for biomass and since only one higher light irradiance was tested, change this statement to 'a higher light irradiance'.

Line 231. Change to 'the first round of restreaking.

Line 257. Insert 'the' before 7002.

In Fig. 5 it is hard to see some of the lines which overlap. Could the authors indicate which samples are in the overlapping lines by adding some arrows with labels.

In Fig 6 from a visual inspection of cells in (C) it looks like the highest cell accumulation occurs on day 7. Can the authors suggest a reason for this.

Line 411- 43 is highlighted.

Methods: In terms of media preparation often small details, like the order reagents are added, can make a difference. If the authors have a step by step guide to making media with details on weights to add to different solutions, then adding this to the supp info would be greatly appreciated.

Reviewer #2 (Remarks to the Author):

The authors have isolated and characterised a new marine strain of cyanobacteria called *Synechococcus* sp. PCC 11901. They show that PCC 11901 is fast growing and accumulates exceptional biomass (compared to other fast growing strains) with a newly develop growth medium, is genetically tractable and could be engineered to produce products (in this case free fatty acids) at yields comparable to *E. coli*. This is a thorough and well-constructed manuscript that covers a detailed characterisation of PCC 11901, including comparisons with the model marine species PCC 7002 and previously reported fast growing fresh water strain UTEX 2973. This work will be of interest to a wide range of cyanobacterial researchers and biotechnologists.

I include a few comments/questions below that I would appreciate that the authors address:

Line 80: The E. coli paper referred to FFA production is from 2011. Are there more recent publications with heterotrophic species that the authors can compare to? (i.e. have het species achieved better since then?)

line 98: I could not find the strain at the PCC collection – could the authors provide a link?

Fig S1: gly+ and glu+ should be defined in the legend.

Line 143: "growth rates did not differ significantly in the 0-2.5% (w/v) NaCl concentration". Can you show these stats? Fig. 1b shows a small but possibly sig. increase at ~2%.

Line 149: Is UTEX 2973 reported to show salt tolerance? Perhaps need to rephrase as this could be considered an unfair comparison.

Line 155/Table S1: Statistical analysis for table S1 would be good to support the statements in this paragraph (e.g. ANOVA and Post Hoc).

Line 161: "...at 41 °C and 1% (v/v) CO₂". Light levels should also be reported.

Fig. 5 – I find the headings above each graph ambiguous, as they could be interpreted as supplementation of each mineral alone (as opposed to additive as the legend states i.e. 4c is supplemented with phosphate AND nitrate and iron). The headings should be modified to reflect this more accurately.

Line 286 and Fig. 6 outlines different light intensity regimes for 7942 and 6803 vs 2973 and 11901 during the early growth stages. Can the authors briefly clarify why they chose this approach in the text.

Fig. 6c and 6d are not clearly explained. It's not really clear what 6c is supposed to be showing, and the decline in the PBS peak should be highlighted in 6d and clarified in the text.

Reviewer #3 (Remarks to the Author):

This manuscript reports the isolation, characterization and genetic modification of the cyanobacterium *Synechococcus* PCC 11901. The fast growth, tolerance to salinity, high biomass productivity, and ease of genetic engineering are desirable features in algal biotechnology. This strain has potential to be adapted by other researchers to become another model cyanobacterium. The medium optimization is a good example in algal biotechnology. The work is comprehensive and solid. The writing is concise and clear.

On line 406, the word phosphorous should be phosphorus.

Reviewer #4 (Remarks to the Author):

This is a most interesting ms describing a novel cyanobacterial strains with promising features for biotechnological applications. It is a comprehensive "story", from isolation via identification, characterization and genome analysis towards molecular tools for genetic engineering, improved media and optimal growth and finally a demonstration of significant photoautotrophic production of free fatty acids. Original results that will have an impact in the field. I think the authors may rephrase some words/text regarding the need to supplement the growth media with cobalamin. Being isolated from a marine environment it is not too surprising it requires cobalamin for growth. However, as stated (lines 195-196) two variants of methionine synthase are present in the genome. A potential mutation leading to loss of activity or expression is suggested. Question: Transcribed and translated? Later (lines 413 and forward) the authors discuss that for industrial application this need for cobalamin easily can be overcome by heterologous expression of MetE. Question: Potential genes are present, even an inactive gene in one plasmid (inactive in what sense?) but external cobalamin still needed. On

what base can the authors state that this can easily be overcome by heterologous expression of MetE?

Referee expertise:

Referee #1: Cyanobacteria biotechnology

Referee #2: Cyanobacteria biotechnology

Referee #3: Metabolic engineering in cyanobacteria

Referee #4: Cyanobacteria genetics and metabolic engineering

Reviewers' comments:

Reviewer #1 (Remarks to the Author):

This is a landmark paper and a major advance in developing cyanobacteria as a biotechnology platform. While 'fast growing' cyanobacterial strains, such as *Synechococcus* UTEX 2973, have previously been reported, results from these papers show growth only at early log phase, which is not useful for biotechnology. I commend the authors for performing these studies properly by demonstrating fast growth to high cell concentrations and biomass, and for their laborious experiments on media optimisation. I am happy to support publication of this paper once the following comments have been addressed.

One big issue is how do the authors know that the higher biomass and OD of 11901 compared to the other species is due to faster growth and not differences in cell properties? It could be due to differences in cell size or in their optical properties of the cells (which could be accumulating storage compounds like polyhydroxybutyrate that change these). Were cell counts taken? Can the authors confirm there are more cells in the 11901 cultures at later growth stages?

We agree with the reviewer that OD is not an accurate measurement of cell density and so we have done additional experiments to determine cell number and cell size as a function of OD of the culture during growth. These experiments were performed on cultures of PCC 11901 and 7002 grown for 10 days in MAD2 medium at 5% CO₂ and the results are presented in Supplementary Figures 11-13.

Although the measured OD₇₃₀ and dry cell weight values for both strains differed greatly (nearly 60% higher OD and 37% higher dry cell weight for PCC 11901), we found that the cell count was very similar for both cultures, measured either using a hemocytometer or by FACS. There was no indication using both techniques for the presence of short filaments or doubling cells that might have skewed the data.

In contrast, analysis of cell size by both microscopy imaging and FACS revealed that compared to the 7002 strain, the PCC 11901 cells became much larger after 5 days, which helps explain the observed divergence between cell counts and OD₇₃₀/biomass for PCC 11901 and PCC 7002. These differences in size were statistically analyzed using ANOVA and post-hoc and found to be significant. We have

presented this new information in the revised manuscript in results (2.6) and methods (4.7 and 4.10) sections.

In terms of the possibility that PCC 11901 accumulates polyhydroxybutyrate, analysis of the genome sequence of PCC 11901 indicates that it lacks the *phaA*, *phaB* and *phaC* genes involved in polyhydroxybutyrate synthesis, which would indicate that this particular molecule is not used as a storage compound.

Minor comments

The supplementary labels are not in the correct numerical order within the text (i.e. Fig. S1C is first in the text and fig. S1a or b are not mentioned). Please correct this and check the manuscript thoroughly.

Supplementary Figure 1 corrected.

Fig S5. It is hard to see both the biomass and growth measurements when presented in the same graph. Can this be split into separate graphs.

Supplementary Figure 5 corrected – OD and biomass graphs are now separate.

Line 167. It looks like some of the strains accumulated similar biomass at both light intensities although it is hard to tell since the graphs are so crowded. Please add numbers for biomass and since only one higher light irradiance was tested, change this statement to 'a higher light irradiance'.

The sentence has been rephrased and more details regarding biomass accumulation were added: 'In all light conditions tested, PCC 11901 accumulated more biomass after 4 days of growth (4.9 g/L) than PCC 7002 and UTEX 2973 (3.7 and 2.5 g/L respectively) (**Supplementary Fig. 5a**). However, all strains accumulated slightly less biomass at high light irradiances, an effect especially noticeable in the case of PCC 7002, which decreased from 3.1 to 2.6 g/L at later time points (**Supplementary Fig. 5b**).'

Line 231. Change to 'the first round of restreaking.

Line 229 Corrected to 'the first round of restreaking'.

Line 257. Insert 'the' before 7002.

Correction made in Line 256.

In Fig. 5 it is hard to see some of the lines which overlap. Could the authors indicate which samples are in the overlapping lines by adding some arrows with labels.

Arrows with colors corresponding to different samples have been added in Figure 5.

In Fig 6 from a visual inspection of cells in (C) it looks like the highest cell accumulation occurs on day 7. Can the authors suggest a reason for this.

This figure was included to illustrate the difference in pigmentation for the different strains over time and is not meant to be representative of culture density. The PCC 11901 sample at 7 days was loaded onto the 96 well plate at a 2x lower dilution than samples from other time points, resulting in an apparently higher cell density. Though it may appear that the biomass accumulation was highest on day 7, the OD and biomass accumulation data do not reflect that. The figure legend has been edited to clarify this point.

Line 411- 43 is highlighted.

In the current version of the manuscript the only highlighted text is there to reflect changes to the previously submitted version.

Methods: In terms of media preparation often small details, like the order reagents are added, can make a difference. If the authors have a step by step guide to making media with details on weights to add to different solutions, then adding this to the supp info would be greatly appreciated.

Detailed recipes for AD7, MAD and MAD2 media are now described in the Supplementary material, including order of addition of reagents.

Reviewer #2 (Remarks to the Author):

The authors have isolated and characterised a new marine strain of cyanobacteria called *Synechococcus* sp. PCC 11901. They show that PCC 11901 is fast growing and accumulates exceptional biomass (compared to other fast growing strains) with a newly develop growth medium, is genetically tractable and could be engineered to produce products (in this case free fatty acids) at yields comparable to *E. coli*. This is a thorough and well-constructed manuscript that covers a detailed characterisation of PCC 11901, including comparisons with the model marine species PCC 7002 and previously reported fast growing fresh water strain UTEX 2973. This work will be of interest to a wide range of cyanobacterial researchers and biotechnologists.

I include a few comments/questions below that I would appreciate that the authors address:

Line 80: The *E. coli* paper referred to FFA production is from 2011. Are there more recent publications with heterotrophic species that the authors can compare to? (i.e. have het species achieved better since then?)

To our knowledge the highest FFA yields for *E. coli* (5.2 g/L) have been described in Zhang et al. (doi: 10.1016/j.ymben.2012.08.009), which we have described in Table 2 (line 496). We have already highlighted the highest yields achieved so far by another heterotroph in the first draft of the manuscript and in the revised version between lines 485-487: 'Though these values are much lower than the highest ever reported FFA production titre of 50.2 g/L, using a heavily engineered *Rhodococcus opacus* strain, this latter system requires a high concentration of glucose⁵².'

line 98: I could not find the strain at the PCC collection – could the authors provide a link?

The strain can now be obtained from the PCC collection. For details please refer to the catalogue website: <https://catalogue-crbip.pasteur.fr/>

Fig S1: gly+ and glu+ should be defined in the legend.

Supplementary Fig. 1 legend corrected: ‘Glycerol and glucose tolerance of PCC 11901 and photoheterotrophy analysis. Wild-type (*wt*), glycerol (*gly*⁺) and glucose (*glc*⁺) adapted strains were grown...’

Line 143: “growth rates did not differ significantly in the 0-2.5% (w/v) NaCl concentration”. Can you show these stats? Fig. 1b shows a small but possibly sig. increase at ~2%.

We have performed ANOVA and post hoc statistical analysis of the growth rates at different NaCl concentrations (for details please refer to Supplementary data 2). The results differ depending on how many groups/conditions have been compared in each test. When all conditions were compared, there was no significant difference in growth rate at NaCl concentrations ranging from 0 to 5%.

ANOVA	0-10%	at least 1 condition is different
two-tailed t-Test	0% vs. 1%	no significant difference
	0% vs. 1.8%	no significant difference
	0% vs. 2.5%	no significant difference
	0% vs. 5%	significantly different
	0% vs. 7.5%	significantly different
	0% vs. 10%	significantly different
	1% vs. 1.8%	no significant difference
	1% vs. 2.5%	no significant difference
	1% vs. 5%	significantly different
	1% vs. 7.5%	significantly different
	1% vs. 10%	significantly different
	1.8% vs. 2.5%	no significant difference
	1.8% vs. 5%	no significant difference
	1.8% vs. 7.5%	significantly different
	1.8% vs. 10%	significantly different
	2.5% vs. 5%	significantly different
	2.5% vs. 7.5%	significantly different
	2.5% vs. 10%	significantly different
	5% vs. 7.5%	significantly different
	5% vs. 10%	significantly different
7.5% vs. 10%	no significant difference	

Analysis of the results referring to salinities between 0 and 2.5% NaCl (excluding all others from the t-test analysis) showed that there was no significant difference in the NaCl 0-1.8% range.

ANOVA	0-2.5%	at least 1 condition is different
two -tailed t-Test	0% vs. 1%	no significant difference
	0% vs. 1.8%	no significant difference
	0% vs. 2.5%	no significant difference
	1% vs. 1.8%	no significant difference
	1% vs. 2.5%	no significant difference
	1.8% vs.2.5%	significantly different

Line 149: Is UTEX 2973 reported to show salt tolerance? Perhaps need to rephrase as this could be considered an unfair comparison.

Sentence (line 147 in the revised manuscript) has now been changed to: ‘While PCC 7002 can also grow in the presence of high salt (up to 9% (w/v) NaCl³²), we found that in our hands the PCC 11901 strain exhibited higher salt tolerance (**Supplementary Fig. 3d**).’

Line 155/Table S1: Statistical analysis for table S1 would be good to support the statements in this paragraph (e.g. ANOVA and Post Hoc).

Complete statistical analysis (ANOVA and Post Hoc) for table S1 can be found in Supplementary material 2 and the summary is presented below:

Growth conditions	ANOVA (3 strains)	Doubling time difference		
		two-tailed t-Test		
		11901 vs. 7002	11901 vs. 2973	7002 vs. 2973
41°C, 300 $\mu\text{mol photons}\cdot\text{m}^{-2}\cdot\text{s}^{-1}$, 1%	at least 1 strain is different	no significant difference	significantly different	no significant difference
41°C, 500 $\mu\text{mol photons}\cdot\text{m}^{-2}\cdot\text{s}^{-1}$, 1%	at least 1 strain is different	no significant difference	significantly different	significantly different
38°C, 300 $\mu\text{mol photons}\cdot\text{m}^{-2}\cdot\text{s}^{-1}$, 1%	at least 1 strain is different	no significant difference	significantly different	significantly different
38°C, 500 $\mu\text{mol photons}\cdot\text{m}^{-2}\cdot\text{s}^{-1}$, 1%	at least 1 strain is different	significantly different	significantly different	significantly different
38°C, 660 $\mu\text{mol photons}\cdot\text{m}^{-2}\cdot\text{s}^{-1}$, 1%	at least 1 strain is different	no significant difference	no significant difference	no significant difference
30°C, 300 $\mu\text{mol photons}\cdot\text{m}^{-2}\cdot\text{s}^{-1}$, 1%	at least 1 strain is different	significantly different	significantly different	significantly different
38°C, 100 $\mu\text{mol photons}\cdot\text{m}^{-2}\cdot\text{s}^{-1}$, air	no significant difference	no significant difference	no significant difference	no significant difference
38°C, 300 $\mu\text{mol photons}\cdot\text{m}^{-2}\cdot\text{s}^{-1}$, air	at least 1 strain is different	no significant difference	significantly different	significantly different

Line 161: “...at 41 °C and 1% (v/v) CO₂”. Light levels should also be reported.

We thank the reviewer for spotting the omission. The details have now been updated as follows: ‘Under our growth conditions, UTEX 2973 was the fastest growing strain (shortest doubling time of 1.93 ± 0.04 hours at 41 °C, 500 $\mu\text{mol photons}\cdot\text{m}^{-2}\cdot\text{s}^{-1}$ and 1% (v/v) CO₂)...’

Fig. 5 – I find the headings above each graph ambiguous, as they could be interpreted as supplementation of each mineral alone (as opposed to additive as the legend states i.e. 4c is supplemented with phosphate AND nitrate and iron). The headings should be modified to reflect this more accurately.

We have modified the headings to: ‘AD7, varying NaNO₃’, ‘AD7 +NaNO₃, varying FeCl₃’ and ‘AD7 +NaNO₃ +FeCl₃, varying KH₂PO₄’

Line 286 and Fig. 6 outlines different light intensity regimes for 7942 and 6803 vs 2973 and 11901 during the early growth stages. Can the authors briefly clarify why they chose this approach in the text.

The rationale behind using this light regime is explained in the materials and methods section (line 564-566: 'In the case of PCC 7942 and PCC 6803, due to their lower light tolerance in dilute cultures, the initial light intensity was set to $75 \mu\text{mol photons}\cdot\text{m}^{-2}\cdot\text{s}^{-1}$, then changed to $150 \mu\text{mol photons}\cdot\text{m}^{-2}\cdot\text{s}^{-1}$ after 1 day and further increased to $750 \mu\text{mol photons}\cdot\text{m}^{-2}\cdot\text{s}^{-1}$ the next day.'

Initially we tried to grow all strains at higher starting light intensity ($150 \mu\text{mol photons}\cdot\text{m}^{-2}\cdot\text{s}^{-1}$), but few replicates of PCC 6803 and 7942 cultures bleached.

Fig. 6c and 6d are not clearly explained. It's not really clear what 6c is supposed to be showing, and the decline in the PBS peak should be highlighted in 6d and clarified in the text.

Figure 6c shows physical appearance (pigmentation) of the cultures, the easiest measure of a fitness of the culture. It was meant to show that the cultures of 2973 and 6803 appear more stressed (pale green) compared to other strains from day 5 and 8 onwards respectively. The position of the major absorbance bands for the main pigments have now been highlighted in Fig. 6d.

Phycobilisome degradation has been already described in the first draft of the manuscript in lines 294-296 (289-292 in the revised manuscript): 'Regarding culture fitness, loss of the light-harvesting phycobilisome complex, which is symptomatic of general stress⁴³ was already apparent in the case of UTEX 2973 after 3-4 days of cultivation but less apparent in the PCC 7942 and 11901 strains, even after 10 days of cultivation (Fig. 6c-d).'

Reviewer #3 (Remarks to the Author):

This manuscript reports the isolation, characterization and genetic modification of the cyanobacterium *Synechococcus* PCC 11901. The fast growth, tolerance to salinity, high biomass productivity, and ease of genetic engineering are desirable features in algal biotechnology. This strain has potential to be adapted by other researchers to become another model cyanobacterium. The medium optimization is a good example in algal biotechnology. The work is comprehensive and solid. The writing is concise and clear.

On line 406, the word phosphorous should be phosphorus.

We thank the reviewer for spotting the typo. We have corrected the sentence to: 'It was recently shown that engineering cyanobacteria to utilize unconventional **phosphorus** and nitrogen sources can dramatically reduce the risk of contamination'

Reviewer #4 (Remarks to the Author):

This is a most interesting ms describing a novel cyanobacterial strains with promising features for biotechnological applications. It is a comprehensive "story", from isolation via identificaton, characterization and genome analysis towards molecular tools for genetic engineering, improved media

and optimal growth and finally a demonstration of significant photoautotrophic production of free fatty acids. Original results that will have an impact in the field. I think the authors may rephrase some words/text regarding the need to supplement the growth media with cobalamin. Being isolated from a marine environment it is not too surprising it requires cobalamin for growth. However, as stated (lines 195-196) two variants of methionine synthase are present in the genome. A potential mutation leading to loss of activity or expression is suggested. Question: Transcribed and translated? Later (lines 413 and forward) the authors discuss that for industrial application this need for cobalamin easily can be overcome by heterologous expression of MetE. Question: Potential genes are present, even an inactive gene in one plasmid (inactive in what sense?) but external cobalamin still needed. On what base can the authors state that this can easily be overcome by heterologous expression of MetE?

We agree with the reviewer that the presence of a plasmid-encoded *metE* gene in PCC 11901 is an interesting finding given that PCC 11901 requires cobalamin for growth.

We have, in response to the reviewer's query, done a complementation experiment to test whether *metE* from PCC 11901 is functional in PCC 7002. We cloned a DNA fragment encompassing the predicted cobalamin riboswitch and *metE* gene of PCC 11901 into a plasmid carrying the *glpK* site of PCC 7002. Then we transformed PCC 7002 with the construct and plated it on AD7 agar plates without cobalamin. Unfortunately, despite incubating plates for nearly 2.5 weeks, no colonies appeared even at 1% CO₂ (usually in these conditions transformants appear after 4-5 days). This preliminary data may suggest that either the B12 riboswitch and/or the *metE* gene from 11901 are not functional or poorly functional (at least in PCC 7002). Given this, considerably more work will be needed to investigate the functionality of *metE* in PCC1, such as analyzing transcription and translation as suggested by the reviewer. However, we feel that such a detailed functional analysis of this gene is beyond the scope of the present study.

Given that we cannot exclude some residual activity, we have revised the text to make it clear that MetE in PCC 11901 might be inactive or insufficiently active for rapid growth (lines 193-196 and 420-422).

Regarding heterologous expression of MetE, we were referring to the work of Perez et al. (doi: 10.1128/JB.00475-16), in which expression of a non-B12 requiring MetE enzyme (either from PCC 73109 or another organism) could be used to generate a B12-independent 11901 strain (lines 418-420).

REVIEWERS' COMMENTS:

Reviewer #1 (Remarks to the Author):

All my comments have been addressed and I am happy for this manuscript to be published.

Reviewer #2 (Remarks to the Author):

I thank the authors for responding to my comments. I am happy with the responses and amendments made, both to myself and the other reviewers.

Reviewer #4 (Remarks to the Author):

Revised version ok.